# Directly imaging excited state-resolved transient structures of water induced by valence and inner-shell ionisation

Zhenzhen Wang [1,7], Xiaoqing Hu [2,7], Xiaorui Xue [3,7], Shengpeng Zhou[1], Xiaokai Li[1], Yizhang Yang[1], Jiaqi Zhou [3], Zheng Shu[2], Banchi Zhao[1], Xitao Yu [1], Maomao Gong[4,5], Zhenpeng Wang[2,4], Pan Ma[1], Yong Wu [2,6] ✉, Xiangjun Chen[4], Jianguo Wang[2], Xueguang Ren[3] ✉, Chuncheng Wang [1] ✉ & Dajun Ding [1] ✉

Real-time imaging of transient structure of the electronic excited state is fundamentally critical to understand and control ultrafast molecular dynamics. The ejection of electrons from the inner-shell and valence level can lead to the population of different excited states, which trigger manifold ultrafast relaxation processes, however, the accurate imaging of such electronic state-dependent structural evolutions is still lacking. Here, by developing the laser-induced electron recollision-assisted Coulomb explosion imaging approach and molecular dynamics simulations, snapshots of the vibrational wave-packets of the excited (A) and ground states (X) of $D_2O^+$ are captured simultaneously with sub-10 picometre and few-femtosecond precision. We visualise that $\theta_{DOD}$ and $R_{OD}$ are significantly increased by around 50° and 10 pm, respectively, within approximately 8 fs after initial ionisation for the A state, and the $R_{OD}$ further extends 9 pm within 2 fs along the ground state of the dication in the present condition. Moreover, the $R_{OD}$ can stretch more than 50 pm within 5 fs along autoionisation state of dication. The accuracies of the results are limited by the simulations. These results provide comprehensive structural information for studying the fascinating molecular dynamics of water, and pave the way towards to make a movie of excited state-resolved ultrafast molecular dynamics and light-induced chemical reaction.

Capturing the transient position of nuclei with sub-Angstrom and femtosecond spatiotemporal resolution for a specific electronic state can visualise the ultrafast structural dynamics of polyatomic molecules, which is a key step towards a better understanding and controlling of the chemical reaction. Many ultrafast structural imaging techniques have been developed to achieve this goal, such as diffraction imaging techniques with X-ray[1,2] and ultrafast electron bunch[3–6], laser-induced electron diffraction (LIED)[7–10] and Coulomb explosion imaging (CEI)[11,12]. Using LIED, Blaga et al. and Wolter et al. established that the extension of carbon–hydrogen bond occurs at a time scale of a few femtoseconds, suggesting that femtosecond to sub-femtosecond time resolutions are critical for capturing the

[1]Institute of Atomic and Molecular Physics and Jilin Provincial Key Laboratory of Applied Atomic and Molecular Spectroscopy, Jilin University, 130012 Changchun, China. [2]Key Laboratory of Computational Physics, Institute of Applied Physics and Computational Mathematics, 100088 Beijing, China. [3]School of Physics, Xi'an Jiaotong University, 710049 Xi'an, China. [4]Hefei National Research Center for Physical Sciences at Microscale and Department of Modern Physic, University of Science and Technology of China, 230026 Hefei, China. [5]School of Physics and Information Technology, Shaanxi Normal University, 710119 Xi' an, China. [6]HEDPS, Center of Applied Physics and Technology, Peking University, 100871 Beijing, China. [7]These authors contributed equally: Zhenzhen Wang, Xiaoqing Hu, Xiaorui Xue. ✉e-mail: wuyong@iapcm.ac.cn; renxueguang@xjtu.edu.cn; ccwang@jlu.edu.cn; dajund@jlu.edu.cn

motion of hydrogen in electronic excited states[8,9]. The CEI was developed decades ago to study the molecular structure by ionising the target to a highly charged state upon various radiation sources, such as X-ray, electron beams and intense femtosecond laser pulses. The molecular structure prior to Coulomb explosion (CE) can be reconstructed from the momentum information of all the fragments in the molecular frame of reference[11]. Benefited from the ion momentum imaging technique, CEI has equal sensitivity for the probing of light and heavy atoms[13], and has been used to trace the roaming dynamics of hydrogen within the molecules[14] and the conformation of the chiral molecule and complex organic molecule[11,13]. However, retrieval of the accurate electronic-state resolved structure of molecules with CEI is still elusive, since the complex nuclear dynamics during the multiple ionisation can distort the mapping between the initial structure to the observed momentum information[15]. For the sequential ionisation of valence electrons driven by the femtosecond laser pulse, the bond length of molecules retrieved from the CEI is usually larger than the equilibrium value, which can be explained by a concerted enhanced ionisation model that the ionisation yields reach maximum as the bond length extends to a critical value in the cation or dication states[16]. To capture the accurate transient position of the atom including hydrogen with CEI, one needs to freeze the nuclear motion during the multiple ionisation by projecting the molecule to the higher charge state within the few-femtoseconds regime. For the CEI of polyatomic molecules induced by electron impact, ultrafast nuclear relaxation dynamics can occur after inner-shell ionisation, and its structure evolution closely relates to the time scale of Auger decay. Moreover, a direct comparison study of the CEI induced by the ionisation of intense laser pulses and electron impact can provide solid experimental evidence to reveal the electronic state-resolved ultrafast nuclear relaxation dynamics of polyatomic molecules.

The structural deformation of water triggered by ionising radiation is essential for many research areas, ranging from radiation damage of biological tissue to the complex water-related reactions in interstellar space and the production of $H_3O^+$ and OH radicals[17–19]. Significant efforts have been devoted to image the transient structure of the water molecule[20–22]. Légaré presented a symmetrically stretched and straightened geometry of water with a few-cycle intense laser pulse based on CEI[23]. Liu et al. applied LIED to image the water molecule and observed the most probable structure of ionic states with a stretched bond length[24]. Interestingly, the electron can be tunnelled from both the highest occupied molecular orbital (HOMO) and HOMO-1 of water, which can initiate dramatically different vibrational wave-packet evolutions and lead to the population of the ionic ground (X) and excited states (A)[25]. By ionising inner-shell electron with X-ray free electron laser followed by Auger decay, the bond stretching at different instants along dication is probed with CEI, where the structure relaxation during the Auger decay (assumed to be 3 fs) is ignored[20]. Thus far, direct imaging of the excited-state resolved nuclear motions of water and comparing the structure deformation induced by inner-shell and valence ionisation remains largely unexplored.

In this work, by developing the laser-induced electron recollision-assisted Coulomb explosion imaging approach, we reveal the ultrafast nuclear relaxation along the ground and excited electronic states of cation and dication, and retrieved the accurate geometry of neutral $D_2O$. With the help of the simulations, the snapshots of the vibrational wave-packets of the excited (A) and ground states (X) of $D_2O^+$ are captured simultaneously with sub-10 picometre and few-femtosecond spatiotemporal precision. We show the $R_{OD}$ can stretch more than 50 pm within approximately 5 fs after double inner-shell ionisation induced by electron impact. The accuracies of those results are limited by the simulations.

## Results

### Imaging of the excited states-resolved structures triggered by the valence ionisation

Experimentally, $D_2O$ is irradiated by linearly and circularly polarised intense laser pulses leading to the removal of three outer-valence electrons of the molecule, which is followed by the three-body CE process $D_2O^{3+} \longrightarrow D^+ + O^+ + D^+$. Measurement of the three-dimensional momenta of the fragment ions that are dissociated from $D_2O^{3+}$ enables the kinetic energy release (KER, sum energy of three fragments) and the angle between the two momentum vectors of $D^+$ ($\theta_{cm}$) in the centre-of-mass frame to be derived. The measured density plots of $\theta_{cm}$ versus KER are compared between linear and circular polarisations, which are shown in Fig. 1a and b, respectively. The plots exhibit distinctive features including the low- and high-KER region (labelled region I and II, respectively). The region I (centred at ~20 eV) contains three components with different $\theta_{cm}$. In the case of an circularly polarised laser field (Fig. 1b), the distributions are nearly identical, which shows weak polarisation dependence. Those events originate from the sequential triple ionisation via the well-known strong field enhanced ionisation (EI) mechanism, where ultrafast bond extension within a few fs along the potential energy surface (PES) of the dication leads to the decrease of the KER[16,22]. We proposed a three-body electron recollision-assisted Coulomb explosion (TERCE) approach, where the sub-cycle laser-induced electron recollision process is used to freeze the nuclear motion in the EI process, leading the production of high-KER region (II). The tunnelling electron from the cation can be driven back to the vicinity of the dication by the oscillating electric field and then further ionise or excite the parent ion before prominent nuclear motion which can occur within one optical cycle (~2 fs in the present case)[26]. This idea has been demonstrated to be available for the proton wave-packet dynamics of $H_2$ molecule[27,28]. TERCE thus can achieve the few-fs temporal resolution. Since the circularly polarised light suppresses the electron recollision dynamics, and region II is expected to be absent in the case of circularly polarised light, which can be seen in Fig. 1b. Moreover, the region II exhibits different maxima of the ($\theta_{cm}$-KER) distributions relative to that originating from the enhanced ionisation process (region I). Furthermore, a comparison between Fig. 1a, b exhibits strong laser polarisation dependence in the region (labelled III) with KER around 23 eV and small $\theta_{cm}$ (less than 70°), which indicates that TERCE plays a key role in its formation.

To further reveal the dissociation dynamics of TERCE, we present the energy-correlations map between two $D^+$ and Dalitz diagrams in Supplementary Fig. 1. Three regions can be well resolved in the energy-correlation maps, where regions II and III appear only in the results of the linearly polarised light. Region II has larger energy than I, and for both regions, two $D^+$ ions have a similar energy, and thus a concerted three-body CE with symmetric geometry can be inferred. However, for region III, the two correlated $D^+$ have large energy differences, suggesting the asymmetric concerted three-body CE process is dominant. These arguments can be further validated by the Dalitz diagram shown in Supplementary Figs. 1e, f, region II appears at the central zone, which stands for the symmetric CE process, and region III shows two arm-like distributions, indicating the asymmetric three-body CE process. By clarifying the concerted three-body CE mechanism for regions II and III, we can further retrieve the transient structure before the three-body CE.

Region II in Fig. 1a contains several components with different peak positions of the ($\theta_{cm}$-KER) distribution, indicating that multiple dissociative pathways are involved. To isolate those pathways, the detailed information of each component needs to be extracted. We first zoom in on the distributions from TERCE by selecting events in high-KER region. In the concerted symmetric CE, the events with the higher KER will have larger energy difference between $D^+$ and $O^+$, which is shown in Supplementary Fig. 2. By selecting the events with energy differences larger than 12 eV, we can keep the events from symmetric

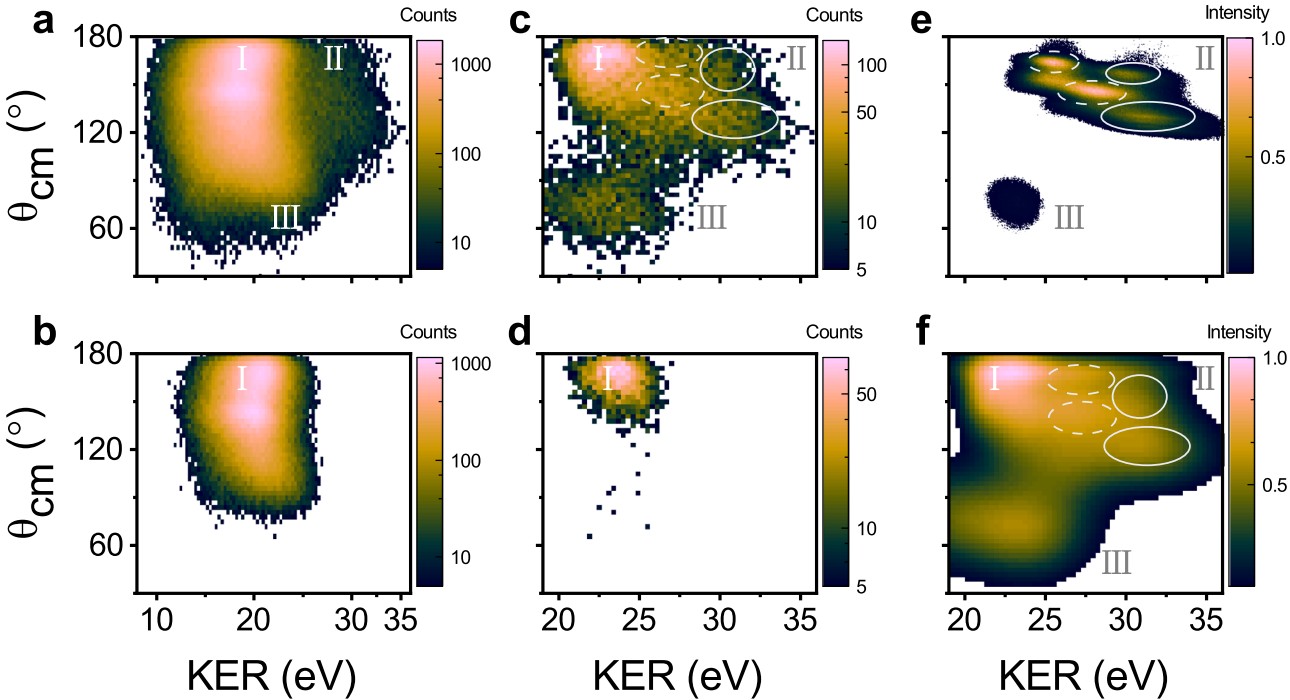

**Fig. 1 | Experimental results and simulations of TERCE. a, b** present the $\theta_{cm}$-KER distributions of the three-body CE channel induced by linearly and circularly polarised 40 fs laser pulses, KER and $\theta_{cm}$ stand for the total kinetic energy release and the bond angle. Here, I labels the enhanced ionisation with weak polarisation independence, and II and III label the distributions with strong polarisation dependence, indicating that these events are correlated with electron recollision processes. **c** $\theta_{cm}$-KER distribution mainly originates from TERCE for linearly polarised laser pulses. The events with larger energy differences (>12 eV) between D⁺ and O⁺ are selected for region I and II. The events from asymmetric CE are added for region III. **d** $\theta_{cm}$-KER distribution for circularly polarised laser pulses. The same conditions as **c** are used. The two-dimensional global fit of (**c**) with seven dominant ($\theta_{cm}$-KER) components are shown in (**f**). The four components in region II are marked by the dashed and solid circles in (**c**) and (**f**). The intensity is plotted on a logarithmic scale. **e** Simulated $\theta_{cm}$-KER distributions based on TERCE. For region II, the $\theta_{cm}$-KER distributions dissociated along the doublet and quartet states of the trication are shown and marked with the dashed and solid circles, respectively. For region III, the $\theta_{cm}$-KER distribution dissociated along the quartet state of trication is shown.

CE in region II and remove most of the events from regions I and III. The results are shown in Fig. 1c. Moreover, the events from the asymmetric TERCE channel (region III) have larger energy differences between two D⁺ (see Supplementary Figs. 1a, c), based on this feature, the events from region III can be selected and added to Fig. 1c. With the same conditions for the circularly polarised light, we show those events in Fig. 1d, and only a small fraction of region I can be observable. With the custom-developed two-dimensional fitting program, we conducted a global fit for Fig. 1c and the fitting output is shown in Fig. 1f. In this fit, only the number of components was assumed, and the peak positions of ($\theta_{cm}$-KER) for each component were not fixed (see Supplementary Fig. 3). The fit is in good agreement with the raw results. For region II, four dominant components can be obtained from the fit, and the peak values of ($\theta_{cm}$-KER) are listed in Table 1. Region III contains one dominant component, which will be discussed in the

following section. A custom-developed molecular dynamic method is used to simulate the TERCE channels, and the results are presented in Fig. 1e, which can quantitatively reproduce the experimental patterns shown in Fig. 1c.

The detailed triple ionisation mechanism mediated by the laser-assisted electron-recollision is sketched in Fig. 2a for region II. Here, electron-recollision-induced double ionisation from the cation to the trication is expected, as indicated by the black arrow in Fig. 2a, because the high-KER events can only be produced when the ultrafast nuclear motions along the PES of the dication are nearly frozen[29]. In the rising edge of the laser pulses, the X and A states are firstly populated by the direct tunnelling ionisation of HOMO and HOMO-1 of neutral water. Interestingly, low vibrational states are activated for the X state and highly excited vibrational states are activated for the A state after the vertical ionisation; thus, the evolutions of the vibrational wave-packet are expected to differ markedly[25,30]. We calculated the time-dependent quantum wave-packet evolutions ionised from the X and A states to the trication as shown in the Fig. 2b, respectively[31]. For the X state, the wave-packets oscillate in a small range, while for the A state the large amplitude vibrations proceed rapidly along both $\theta_{DOD}$ and $R_{OD}$, and the wave-packet distributions at 1, 3, and 5 optical cycles after initial ionisation are shown in P₁₃, P₁₄ and P₁₅, respectively (see Supplementary Figs. 7, 8). The coupling of the X and A state via dipole transition can play a minor role in the TERCE, which will be discussed in the future. We further calculate the most probable instant of the tunnelling ionisation for both X and A states based on the molecular Ammosov−Delone−Krainov (MO-ADK) model[32]. The double ionisation yield via sub-cycle tunnelling depends on both the amplitude of the electric field within the laser pulse and the ionisation potential of the water cation at the ionisation instant. The ionisation potential between

**Table 1 | Comparison of peak positions between experimental fit and simulations**

| States | | Experiment fit | | Simulations | |
|---|---|---|---|---|---|
| Ca. | Tri. | $\theta_{cm}$ (°) | KER (eV) | $\theta_{cm}$ (°) | KER (eV) |
| X²B₁ | ⁴A″ | 120 ± 4 | 31.4 ± 0.7 | 130 | 31.3 |
| | ²A′ | 150 ± 6 | 26.5 ± 0.9 | 145 | 28 |
| A²A₁ | ⁴A″ | 154 ± 8 | 29.8 ± 1.0 | 157 | 29.6 |
| | ²A′ | 170 ± 3 | 25.6 ± 0.5 | 162 | 25.5 |

Peak values of ($\theta_{cm}$-KER) for the four dominant components from the fit of the measurements. The fitting errors are given. These values are assigned to the CE along either the quartet or doublet states of the trication (Tri.), and these components are ionised from the X and A states of the cation (Ca.). The simulated peak values of ($\theta_{cm}$-KER) are also shown for each channel.

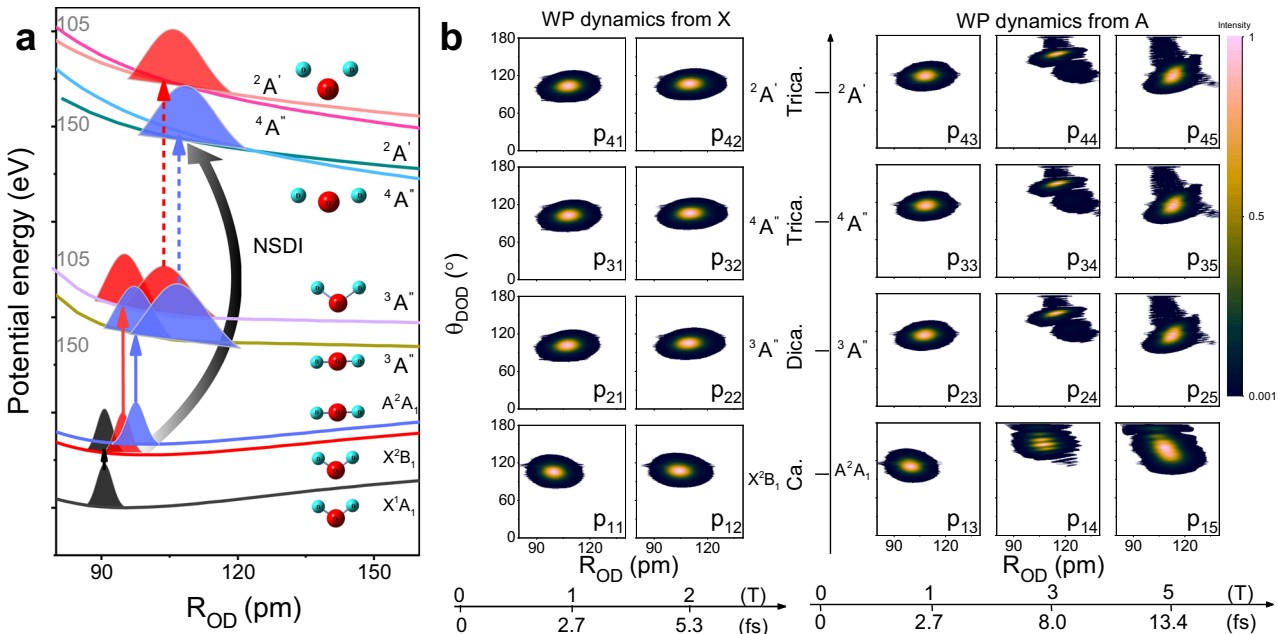

**Fig. 2 | Scheme of TERCE and the time-dependent wave-packet motions.**
**a** Detailed scheme of the TERCE. The electron is firstly ionised to the cation state via tunnelling of the HOMO and HOMO-1 of neutral water, which directly populates the X and A states. With time intervals of a few femtoseconds, the second electron is released via further tunnelling ionisation of the X and A states, and then recollides with the dication and kicks out the third electron, which can be called non-sequential double ionisation (NSDI, indicated as black arrow). Then the trication can dissociate along both the doublet and quartet trication states. The wave-packet evolutions in different states are illustrated as Gaussian distributions. The PESs for 150° are offset for the clarity of presentation, and 150° is the most probable bond angle for the tunnelling ionisation of A state. **b** Simulated time-dependent

vibrational wave-packets dynamics during TERCE. The X axis stands for the time-delay between single ionisation and double ionisation. T stands for the optical cycle (2.7 fs in our case). The panels $P_{11}$ to $P_{15}$ present the vibrational wave-packet (WP) distributions along the X and A states of cation at different time intervals, and are initiated by the tunnelling ionisation of HOMO and HOMO-1. For typical time intervals, the projected wave-packets on dication states are also presented after 2 fs evolutions along the dication (panels $P_{21}$ to $P_{25}$). The panels $P_{31}$ to $P_{35}$ and $P_{41}$ to $P_{45}$ present their corresponding wave-packets before the CE along the doublet and quartet states of the trication, respectively. The electron collision-induced ionisation yields of wave-packets are considered.

the water cation and dication decreases dramatically as the bond length and bond angle increase. As sketched in Fig. 2a, the ionisation energy of dication and trication is lower for larger $\theta_{DOD}$ (150°) than the equilibrium bond angle (105°). Given our experimental conditions, the calculated results reveal that the most probable instant of double ionisation for the X and A states occur approximately 2 and 3 optical cycles after single tunnelling ionisation, respectively (see Supplementary Figs. 9, 12). The distributions of the transient wave-packets at these instants are presented in $P_{12}$ to $P_{14}$ of Fig. 2b, and 150° is the $\theta_{DOD}$ of A state at the most probable double ionisation instant.

The wave-packets at different instants are projected to the dication, and these projected wave-packets further evolve along the PES of the dication within 2 fs (3/4 of one optical cycle) during the electron recollision process[26], and the wave-packet distributions after evolvement are presented in $P_{21}$ to $P_{25}$ of Fig. 2b. The wave-packets are further projected to the dissociative trication state via the electron recollision process, and the final wave-packets distributions are calculated with multicenter three-distorted-wave (MCTDW) method[33]. In those calculations, two assumptions are used as follows: first, the ionisations via strong field tunnelling and electron recollision satisfy the Franck-Condon principle, namely the nuclear vibrational wave-packets stay the same during the ionisation; second, the laser field can be considered as a perturbation interaction, which only affects the orientational distribution and has no influence to the evolutions of vibrational wave-packet along the PESs. Here, the high-precision three-dimensional PESs of the trication state of water were calculated using the multireference configuration interaction (MRCI) method[34,35], and we found that the two lowest-lying states of $D_2O^{3+}$ ($^2A'$ and $^4A''$) have similar potential energies in the 100–130 pm region. Hence, their contributions to TERCE should both be considered here. Based on the

calculated wave-packets distributions on the doublet and quartet states, we simulated the dissociation dynamics along the accurate PESs, and the results are shown in Fig. 1e. All the details about the calculation method are documented in the Supplementary information (see Supplementary Notes 3–5).

The simulated $\theta_{cm}$-KER distributions of the four CE channels ionised from X and A states are presented in region II in Fig. 1e (marked by dashed and solid curves), which well reproduced the $\theta_{cm}$-KER distributions from TERCE in Fig. 1c. The peak values of these four channels are compared quantitatively with the fitted values in Table 1. The quantitative agreement between the simulations and experimental fit verifies that the four components in region II correspond to the four channels starting from the X or A states and ending with the doublet and quartet states of the trication. The simulated peak positions of KER and $\theta_{cm}$ were in good agreement with the two channels from the A state and the channel from the X state (quartet dissociative channel). The deviation is relatively larger for the doublet channel from the X state because its lower-energy distribution in the measurement overlaps with the enhanced ionisation region. Our simulations considered the time-dependent vibrational wave-packet propagation in different ionisation stages, the dependence of the molecular tunnelling ionisation yields on the transient structure including orientation effect, and the electron-recollision induced ionisation yields, and thus enabled the direct comparison and interpretation of experimental results from TERCE.

We attempted to determine the transient geometrical structure based on TERCE. Initially, we assumed that the PES of the trication to be purely Coulomb repulsive[36,37], and the reconstructed O-D bond length ($R_{OD}$) corresponding to the four components are listed in Table 2. Overall, the extracted values are much larger than that of the

**Table 2 | Summary of the retrieved structures of X and A states**

| States | Equilibrium | | States | Coul. approx. | This work | | Simulation | |
|---|---|---|---|---|---|---|---|---|
| Ca. | $\theta_{DOD}$ (°) | $R_{eq.}$ (pm) | Tri. | $R_{Coul.}$ (pm) | $\theta_{DOD}$ (°) | $R_{OD}$ (pm) | $\theta_{DOD}$ (°) | $R_{OD}$ (pm) |
| $X^2B_1$ | 110 | 101 | $^4A''$ | 126 | $97^{+8}_{-6}$ | $108^{+3}_{-2}$ | 106 | 110 |
| | | | $^2A'$ | 136 | $113^{+14}_{-16}$ | $113^{+3}_{-3}$ | 106 | 110 |
| $A^2A_1$ | 180 | 99 | $^4A''$ | 120 | $148^{+12}_{-11}$ | $116^{+4}_{-4}$ | 149 | 115 |
| | | | $^2A'$ | 140 | $159^{+7}_{-6}$ | $116^{+3}_{-3}$ | 149 | 115 |

Retrieved and simulated O–D bond length ($R_{OD}$) and bond angle ($\theta_{DOD}$) of the X and A states. The retrieved bond length based on the Coulomb approximation ($R_{Coul.}$) and accurate PESs (this work) are also compared. The equation represents the equilibrium structure of the ionic X and A states. The most probable geometries of the X and A states before CE from the simulations are shown in the last two columns. The propagated fitting errors are given.

equilibrium values of the neutral ground state (105°-97 pm, $\theta_{Neq}$-$R_{Neq}$). Under this assumption, the two observed CE channels from the same cation state (X or A state) are thought to originate from different initial structures, which is not the case here. To overcome this problem, the calculated high-precision three-dimensional PESs of the trication states were adopted to determine the initial structure before CE. We first searched for geometries that can produce the observed KER based on nuclear dynamics simulation along the calculated PESs; then, starting with these geometries, the three-body CE was simulated along the PESs using the semiclassical trajectory approach. Finally, a comparison of the momentum angle $\theta_{cm}$ between the simulated and experimental results enabled us to derive the transient structures of the X and A states of $D_2O^{3+}$ before CE. The geometries obtained from different channels are listed in Table 2. Importantly, the two channels from the doublet and quartet states of the trication provide an in situ self-reference for the accuracy of structural retrieval. For the two channels starting from the X state and ending to the doublet and quartet trication states, the retrieved geometries of ($\theta_{DOD}$-$R_{OD}$) before CE are (113°-113 pm) and (97°-108 pm), respectively. The excellent agreement between them indicates that the sub-10 pm and few-degree accuracy are attainable with the present TERCE approach. Moreover, the most probable geometry before CE from the simulation is (106°-110 pm), which confirms the accuracy of our structural retrieval. Compared with the equilibrium structure of the neutral ground state, the retrieved $R_{OD}$ before CE extends approximately 11–16 pm during the ionisation process from the neutral to the trication state, and the $\theta_{DOD}$ stays similar to $\theta_{Neq}$ within the error bars. Using accurate transient structural imaging, we can directly establish that, by removing one electron from the HOMO, only large amplitude symmetric bond stretching motion is initiated and the bending motion is rarely activated. In the simulation, the corresponding most probable instant for projecting the wave-packets from the X state to the dication is approximately 2 optical cycles (5 fs) after tunnelling ionisation of the HOMO. The time-dependent wave-packet dynamics are presented in Fig. 2b. More specifically, the wave-packets simulations further show that $R_{OD}$ extends approximately 7 pm within 5 fs in the X state, and further extends 5 pm along the dication within 2 fs during the electron recollison process (see Supplementary Fig. 9). We succeeded in capturing the transient structure of X state for water cation with an unprecedented accuracy (sub-10 pm and few-degree).

For the A state, the extracted transient $\theta_{DOD}$ before CE are 159° and 148° from the doublet and quartet dissociation channels with TERCE, respectively, which are approximately 50° larger than the $\theta_{Neq}$. This indicates that highly excited bending motions are initiated by the direct tunnelling ionisation of HOMO-1. The retrieved $R_{OD}$ before CE for two channels is both 116 pm, which is approximately 20 pm larger than $R_{Neq}$. In our simulation, the most probable structure before CE is (149°-115 pm), which is in quantitative agreement with the

measurement within the error bars for both channels. Moreover, the most probable instant for the double ionisation is 3 optical cycles (8 fs) after the first tunnelling ionisation, and this transient wave-packet distribution has a maximum with the geometry of (146°-107 pm), as shown in Fig. 2b. The value of $\theta_{DOD}$ increases approximately 40° and $R_{OD}$ stretches approximately 10 pm after evolution in the A state within 8 fs. The $R_{OD}$ further extends 7 pm along the PES of the dication within 2 fs, while the variation of $\theta_{DOD}$ is minor during the electron recollision process (see Supplementary Fig. 9). Here, the captured nuclear motion can serve as a dynamical fingerprint to resolve the ionic ground and excited states.

In contrast to the symmetrical three-body CE of the X and A states, the asymmetrical CE channel is also observed as shown in Fig. 1a (region III). This distribution can be well reproduced with the two-dimensional fit in Fig. 1f. These events are selected by setting the kinetic energy difference of two $D^+$ larger than 6 eV according to their characteristic distribution in region III of Supplementary Figure 1. The one-dimensional distributions of KER and $\theta_{cm}$ for the asymmetric channel are presented in Fig. 3a and b, respectively. The measured KER is fitted with a peak of $22.4 \pm 0.3$ eV, and $\theta_{cm}$ is centred at $76 \pm 4°$. Compared to region II, the dramatically different $\theta_{cm}$-KER distribution indicates the involvement of an additional ionic excited state. The ab initio calculations of the electronic states of the water cation have a higher excited state (C) with a highly asymmetrical structure, where the two O-D bonds are asymmetrically stretched and the two deuterium atoms are located on the same side of the molecule[38]. Based on the calculated PESs of the cation, dication and trication, we simulated the three-body CE process; that is, $(O-D-D)^+ \rightarrow (O-D_2)^{2+} \rightarrow (O^+ + D^+ + D^+)$, where the equilibrium geometry of the C state was the initial geometry with $R_{OD} \sim 227$ pm and 122 pm, and $\theta_{DOD} \sim 0°$. In the simulation, the elongation of the O-D bond along the dication state during the ionisation from dication to trication is dominated by the strong field-enhanced ionisation at a critical distance, where the distance between the two D atoms is assumed to be similar to that of the enhanced ionisation of $D_2^+$ molecule[39]. The simulated KER and $\theta_{cm}$ distributions are presented in Fig. 3a and b (also shown in Fig. 1c, region III), and are in quantitative agreement with the experimental results. Thus, region III can reflect the geometrical structure of the C state. However, the ionisation potential of the C state is more than 10 eV higher than that of the X state, suggesting the direct population of the C state via tunnelling ionisation of the neutral ground state is unlikely. Moreover, considering the strong polarisation-dependent yields of region III, we propose that the electron recollision-induced excitation leads to the population of the C state. The details are sketched in Fig. 3c. The tunnelling electron from the neutral ground state can return to the vicinity of the parent ion and excite the parent ion to the C state via recollision process. After the initial population, the vibrational wave-packet evolves along the PES to the global minimum with an asymmetric O-D-D geometry, where the triple ionisation occurs via further strong field-induced double ionisation. For this state, only one channel from the quartet state is dominant because the ionisation potential of the quartet state is much less than that of the doublet state of the trication for the larger bond lengths. According to the measured KER and $\theta_{cm}$, we determined the initial geometry before the CE by adopting the accurate PES of the trication. The retrieved $R_{OD}$ was $277^{+5}_{-4}$ pm and $122^{+2}_{-2}$ pm for the two O-D bonds with a bond angle of $6^{+1}_{-1}°$. Compared to the equilibrium geometry of the C state, the shorter $R_{OD}$ is identical and the larger $R_{OD}$ is extended by a further 50 pm along the PES of the dication. Ultrafast structural imaging enabled the population of the higher excited state C with an asymmetric structure to be experimentally established. Our studies indicate that the large amplitude bending motion and asymmetric stretching motion of molecular dynamics can be efficiently triggered by an electron collision process.

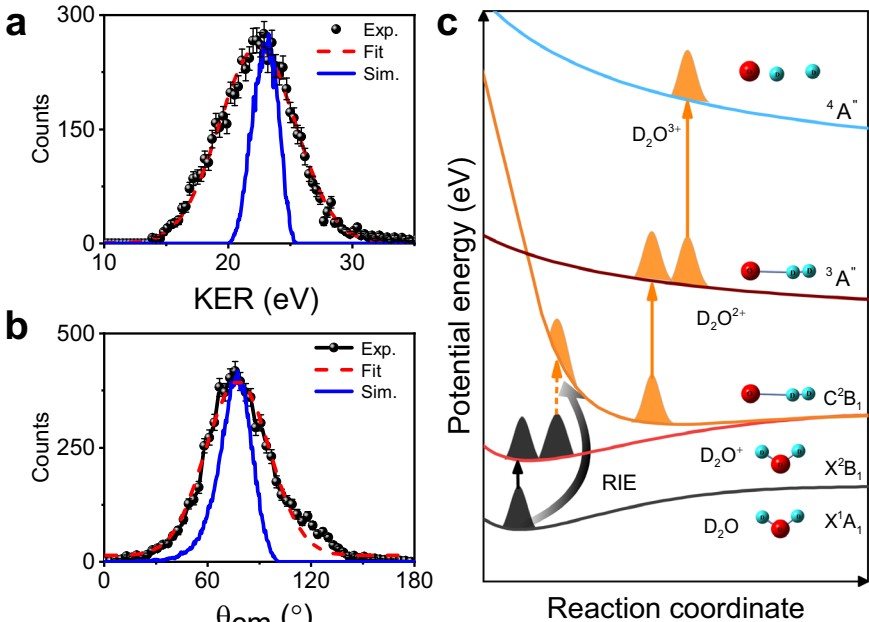

**Fig. 3 | Details of the asymmetric CE channel induced by the electron recollision induced excitation. a**, **b** Measured $\theta_{cm}$ and KER distributions (Exp.) for region III. The dotted red and solid blue lines represent the fit and simulated results (sim.), respectively. **c** Detailed mechanism of the TERCE from the C state. The tunnelling electron is promoted to the ground state of the cation via tunnelling ionisation, and this ionised electron can recollide with the cation and excite a second electron to a higher excited state (C), which is also known as recollision induced excitation (RIE)

(indicated as black arrow). This excitation initiates the large amplitude bending and asymmetric stretching motions. This enables the molecule to relax to the minimum of the PES of the C state, where further tunnelling ionisation to the dication state occurs instantly because of the much lower ionisation potential at larger bond lengths. The wave-packets evolve along the dication for certain time interval and are projected to the trication via enhanced ionisation. Finally, the wave-packet dissociates along the quartet state of the trication.

## Imaging of structure and dynamics triggered by the inner-shell ionisation

The rich nuclei dynamics along the cationic states limit the possibility to retrieve the accurate equilibrium geometry of neutral $D_2O$ experimentally with TERCE, and directly imaging the accurate neutral structure with CEI can serve as a benchmark for the study of water dynamics, which has not been achieved so far. To address this issue, a 200 eV pulsed electron beam is used to induce vertical triple ionisation of $D_2O$, and the three-body CE occurs following the triple ionisation process and the momenta of the produced two $D^+$ and $O^+$ fragments are measured in coincidence. Three electrons from the valence bands can be directly kicked out by the electron impact, and the duration of the electron impact ionisation process is in attosecond regime, which can fully freeze the nuclear motions before CE and enable the accurate imaging of the equilibrium geometry of neutral water with CEI. The measured ($\theta_{cm}$-KER) distribution is shown in Fig. 4a, and three components with highest KER are marked with grey circles. Compared to Fig. 1c, their KERs are larger and $\theta_{cm}$ are all below 150°, indicating the nuclear motions along cationic states in TERCE are frozen by the ultrafast electron impact ionisation. To explore the underlying dynamics of those three components, we implemented a similar molecular dynamics simulation with semi-classical trajectory method as the TERCE, but ignoring all the nuclei motions during the ionisation, i.e., directly projecting the neutral wave-packet to the electronic states of trication, as shown by the black wave-packet in Fig. 4d. Here, three electronic states with the lowest ionisation energy are considered including the doublet and quartet states used in TERCE, and the simulated ($\theta_{cm}$-KER) distributions dissociated along them are shown in Fig. 4b (marked by grey circles). The simulated and measured $\theta_{cm}$ and KER show good agreements, the obtained peak values are presented in Table 3. The consistency confirms that the direct vertical triple ionisation and following CE lead to the production of the three high-energy components, which also verified the validity of our simulation

approach. Based on the measured values of ($\theta_{cm}$-KER) for three pathways, we retrieved the equilibrium geometry with the same procedure used in TERCE, and the obtained ($\theta_{DOD}$-$R_{OD}$) are (102°-94 pm), (100°-99 pm) and (102°-95 pm), respectively. These values are in very good agreement with the theoretical equilibrium (105°-97 pm) for all three independent pathways. These results provide so far the most accurate neutral structure retrieved with the CEI technique experimentally, to the best of our knowledge.

The inner-shell ionisation of polyatomic molecules can also lead to ultrafast nuclei relaxation, which can be different from that induced by the valence ionisation. Our 200 eV electron beam can provide enough energy to kick out two $2a_1$ electrons of $D_2O$, and induce the triple ionisation via the Auger decay process[40,41]. In Fig. 4a, the low-KER component (27.5 eV, marked by red circle) can be observed. The dissociation limit of the lowest quartet state of trication is calculated to be 50.1 eV, and the KER for this pathway is 27.5 eV, thus according to the energy conservation law, the initial energy of autoionisation state should be higher than 77.6 eV, and only the $2a_1^{-2}$ state is in this energy range according to the literature[41]. The most possible pathways to lead to the production of this component are schematically shown in Fig. 4c, two $2a_1$ electrons are firstly ionised by electron impaction and an autoionisation state of dication with the energy of 83.5 eV is populated, which is still lower than the vertical ionisation energy of ground state of trication (85.0 eV)[41]. This can be seen in Fig. 4d, the calculated potential energy curve of the autoionisation state locates below the ground state of trication in the Frank-Condon region. However, as the wave-packet propagates along the stretching motion of $R_{OD}$, the ionisation energy of this state can decrease and cross with the quartet state of the trication, where the triple ionisation via Auger decay is energetically possible. After the crossing point, the Auger decay can occur at certain instant as indicated by the blue arrow, which leads to the low-KER components dissociated along the quartet states. Based on the measured KER and the energy of potential energy curve of

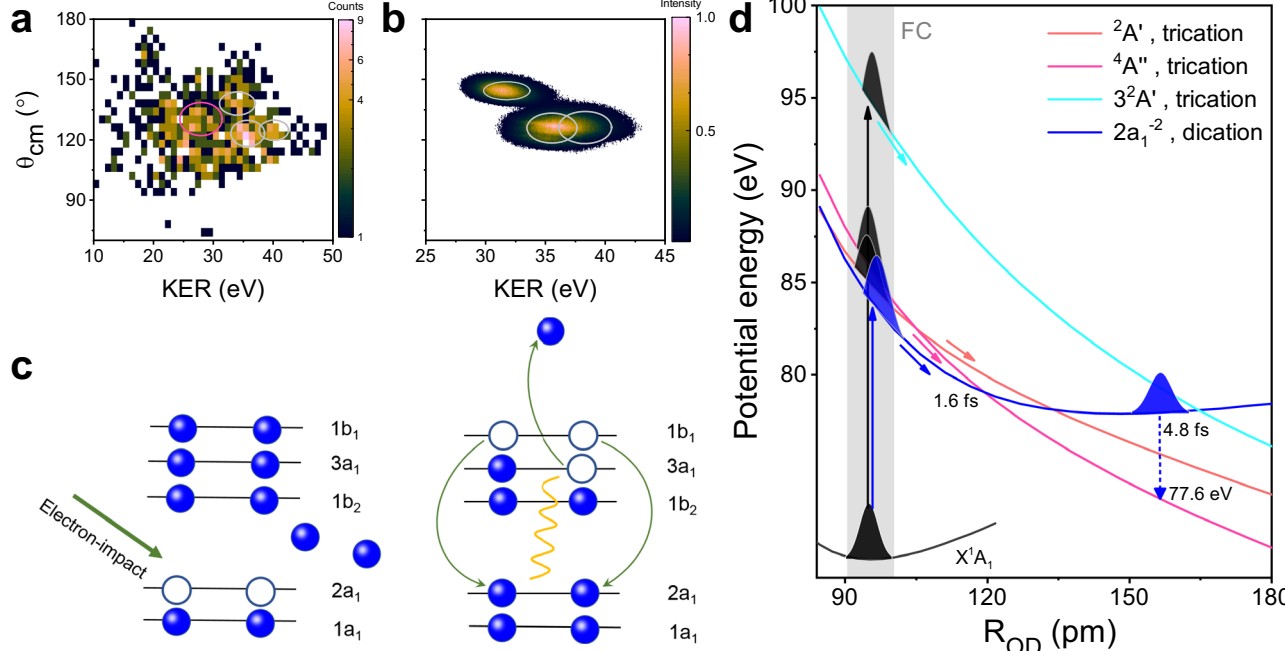

**Fig. 4 | The experimental results induced by the electron impact ionisation.** **a** Measured distribution of ($\theta_{cm}$-KER) for three-body CE induced by the electron impact ionisation. Three components marked with grey solid circles with high-KER originate from the vertical triple ionisation process. One low-KER component marked with red solid circle stand for the three-body CE induced by the inner-shell ionisation process followed by Auger decay. **b** Simulated distributions of $\theta_{cm}$ vs KER for the four components marked in (**a**), which comes from three-body CE induced by the ionisation of both valence and inner-shell electrons. **c** Schematic

plot for the relaxation mechanism for the inner-shell ionisation and Auger decay. Two $2a_1$ electrons are kicked out by the incident electrons, and after the ultrafast nuclear relaxation, two $1b_1$ electrons filled the double holes, and the third electron is ionised by absorbing the radiation. **d** The potential energy curves for the involved electronic states of trication and autoionisation state of dication. The instant of the Auger ionisation is indicated by the vertical blue arrow after ultrafast nuclear relaxation along the autoionisation state.

the quartet state, we can infer the $R_{OD} = 163$ pm at the instant of Auger ionisation for this pathway because of the energy conservation. Based on all these information, we simulated the $\theta_{cm}$ distributions dissociated along this pathway, and the peak value is shown in Table 3. The good agreement of the $\theta_{cm}$ between simulation and measurement confirms the above conclusion. Furthermore, we can infer the time scale of the Auger decay for this pathway based on the measured KER and simulations, which is determined as 3.2 fs. The CEI thus can serve as a clock for the complex Auger decay process in the polyatomic molecule.

**Summary of the electronic state-resolved structural dynamics of water**
In Fig. 5, we summarised the structural evolutions along different electronic states of $D_2O$ induced by the femtosecond laser pulse and electron beam. The equilibrium geometry of $D_2O$ is experimentally

**Table 3 | Strucutres retrieved from the electron impact ioni- sastion of valence and inner-shell electrons**

| States | | Exp. | | Simulation | | Retrieval | |
|---|---|---|---|---|---|---|---|
| neutral | Tri. | $\theta_{cm}$ (°) | KER (eV) | $\theta_{cm}$ (°) | KER (eV) | $\theta_{DOD}$ (°) | $R_{OD}$ (pm) |
| $X^2B_1$ | $3^2A'$ | 127 ± 12.4 | 39.6 ± 0.2 | 127 | 37.8 | $102^{+10}_{-13}$ | 94 ± 1 |
| | $^4A''$ | 122 ± 8.6 | 35.0 ± 0.2 | 126 | 35.6 | $100^{+9}_{-12}$ | 99 ± 1 |
| | $^2A'$ | 144 ± 13.4 | 32.5 ± 0.2 | 144 | 31.5 | $102^{+12}_{-15}$ | 95 ± 1 |
| Dication | | | | | | | |
| $2a_1^{-2}$ | $^4A''$ | 134 ± 21.6 | 27.5 ± 0.2 | 128 | – | 102 | 163 |

Peak values of ($\theta_{cm}$-KER) from the measurement (Exp.) and simulation are listed. The fitting errors are given. The retrieved geometries of neutral heavy water from three high-KER components are listed in the last two rows. The estimated bond length according to the KER is listed for the Auger decay channel.

retrieved from the three-body CE induced by kicking out three valence electrons via electron impact. Ejecting one valence electron through strong field tunnelling leads to the population of X and A state of cation, and based on the simulations, the wave-packets evolve for approximately two and three optical cycles, and their $R_{OD}$ extend 7 pm and 10 pm, respectively. Moreover, the bond angle increases from 97° to 146° for A state. Ejecting the second valence electron at this instant from X and A states lead to the population of the ground state of dication, and according to the retrieved structures, the $R_{OD}$ extends 4 pm and 9 pm within 2 fs, respectively. A higher excited state of cation can be populated by the recollision-induced excitation within 2 fs, and then form an asymmetric geometry before double ionisation, the longer $R_{OD}$ can reach 277 pm before three-body CE. Ionising two inner-shell electrons by electron impact leads to the direct population of autoionisation state of dication, according to our simulations, after the relaxation of 5 fs, the $R_{OD}$ extends approximately 60 pm before CE, indicating the much faster stretching motions along autoionisation state than that for the ground state of dication.

In summary, we captured the remarkable transient geometrical structure for the ionic ground state and excited states of the $D_2O$ molecule at an unprecedented spatiotemporal resolution. We implemented the three-body CEI induced by strong field tunnelling ionisation and direct electron-impact ionisation, which have been well interpreted by our custom-developed 'ab initio' nuclear dynamics simulation. Based on the same observable, the nuclei dynamics induced by the inner-shell and valence electron ionisation are compared. For the cation, we present the bond stretching of 7 pm for the X state within approximately two optical cycles, and significant bending motions (50°) accompanying with 10 pm bond extension within approximately three optical cycles for A state, respectively, according to the comparison between measurements and simulations in the

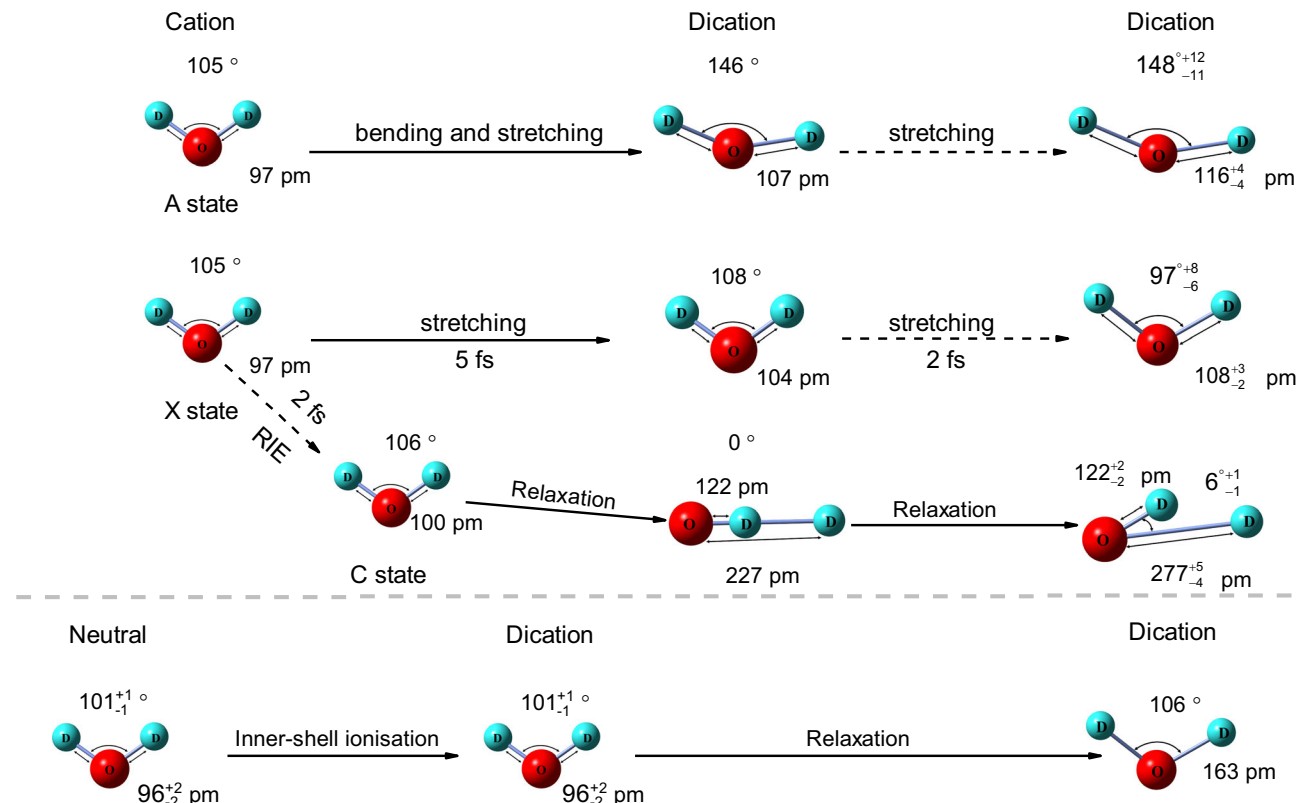

**Fig. 5 | The summary of the structural evolutions for different electronic states of D₂O.** The transient structures at different time instants from the X, A and C states are firstly given based on the simulations in the current condition. For example, the $R_{OD}$ stretches from 97 pm to 107 pm along the A state within approximately three optical cycles, and further extended to 116 pm evolving along the ground state of the dication within 2 fs (electron recollision time). In the lower panels, the retrieved geometry of neutral water on average is 101°-96 pm. The simulations indicate that after ultrafast relaxation within 5 fs along the auto-ionisation state of dication, the $R_{OD}$ stretched to 163 pm for the Auger decay pathway.

present condition. We established that the excited state (C) with O-D-D type structure can be effectively populated by the excitation induced by the recollision process, which may constitute a new possible pathway for the formation of the $H_3O^+$ in interstellar space or proton transfer in the liquid water. For the dication, we visualised the bond extension of 60 pm before Auger decay following the autoionisation state, and this value is much larger than the bond extension along ground state during 2 fs. The accuracies of those results are limited by the simulations. Our results present a comprehensive study of ultrafast transient structure imaging of neutral and ionic water, and the combination of TERCE and CEI induced by electron impact ionisation can provide deep insight into ultrafast nuclear relaxation after ionisation. We demonstrate a robust route for imaging excited-state resolved ultrafast molecular dynamics with sub-10 picometre and few-femtosecond spatiotemporal resolutions.

## Methods

### Experimental details
Laser pulses with a wavelength of 800 nm and pulse duration of 40 fs (full width at half maximum, FWHM) were produced using a Ti-sapphire laser system with a repetition rate of 1 kHz. The laser beam was focused on the molecular beam using a mirror with a focal length of 75 mm. D₂O molecules were evaporated into a chamber using supersonic expansion. All ionic fragments induced by the laser pulses were collected in coincidence using cold target recoil ion momentum spectroscopy (COLTRIMS) (see Supplementary Figure 18), and their three-dimensional momentum vector can be determined by combining the time information of the time-of-flight spectrometer and position information recorded by a position-sensitive detector[42]. The

length of the time-of-flight was 7 cm with the homogenous voltage of 45 Volt/cm. Linearly and circularly polarised laser pulses with the same amplitude of electric field were used for comparison in the experiment, and their peak laser intensities were estimated to be $3.0 \times 10^{14}$ W/cm² and $6.0 \times 10^{14}$ W/cm², respectively, and the corresponding maximum recollision energy of returning electron is approximately 70 eV. The same setup is used for the CEI measurement driven by pulsed electron beam with the energy of 200 eV, and the details have been introduced previously[43-45].

### Theoretical calculations
In this work, a custom-developed molecular dynamic method is used to simulate the three-body breakups of $D_2O^{3+}$. The present simulations consider the complete nuclear wave-packets dynamics for the sequential triple ionisation from D₂O to $D_2O^{3+}$, which enables the direct quantitative comparison with the measurement. Both the changes of vibrational wave-packet and molecular orientation distribution are performed in full quantum time-dependent wave-packet evolution method[31]. The tunnelling ionisations induced by intense laser pulses and the ionisation from dication to trication induced by the electron recollision are considered properly by the semi-classical MO-ADK[32] and Multicenter three-distorted-wave (MCTDW) methods[33], respectively. Detailed theoretical calculation details, calculation steps and all the used approximations can be found in Supplementary Information.

### Reporting summary
Further information on research design is available in the Nature Portfolio Reporting Summary linked to this article.

## Data availability

The data that support the findings of this study are available in Figshare with the identifier https://figshare.com/s/af21f0e747dc1cfd3291[46].

## Code availability

The codes used in this study are available from the corresponding author upon request.

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

## Acknowledgements

This work was supported by the National Natural Science Foundation of China (Grants No. 92261201 (XR, CW), 11934004 (YW), 12134005 (DD), 12274179 (CW), 12104063 (XH) and 12004133 (XL)). We thank Professor Kiyoshi Ueda for the fruitful discussions.

## Author contributions

D.D., C.W., and X.R. conceived the experiments, Z.W., X.X., X.Y., Y.Y., J.Z., B.Z., P.M., and X.L. conducted the measurements, C.W., X.H., X.R, Z.W., and S.Z. interpreted the data, Z.W., X.H. and X.X. prepared the figures. X.H., Z.S., Z.W., M.G., Y.W., X.C., and J.W. provided the calculations. All authors prepared and reviewed the manuscript.

## Competing interests

The authors declare no competing interests.
