## [Peer review file · Nature Communications]

Reviewers' Comments:

Reviewer #1:

Remarks to the Author:

Report on NCOMMS-21-31786E

"Directly imaging excited states-resolved coherent nuclear motions of water with picometer-femtosecond precision" by Zhenzhen Wang et al

In this manuscript, the authors used a laser of 800 nm, 40 fs, 1 kHz, intensity in the range of $3-6 \times 10^4$ W/cm² to multiply ionize D₂O molecules and measured the three-dimensional momentum vectors of the ions, D⁺, D⁺ and O⁺. By analyzing the correlation of the total kinetic energy release (KER) and the angle (θ) between the two D⁺ ions, they obtained the KER- θ plots, for linearly and circularly polarized lights. They were able to distinguish three KER- θ regions (called I, II and III), depending on the high/low KER's and small/large angles. They focused on regions I and II for the D⁺/D⁺/O⁺ breakup of D₂O³⁺ ions, distinguishing that region I is due to symmetric breaking of the two D⁺ ions, while region II is due to the asymmetric breakup. In their model, the D₂O neutral is first tunnel ionized by the laser, creating D₂O⁺ in the ground state X and the first excited state A. The D₂O⁺ is subsequently further doubly ionized by laser-induced electron rescattering to form dissociative D₂O³⁺ ions. Based on this idea and additional assumptions, they stated that the experimental results could be attributed to the breakup of D₂O³⁺ ions at sub-cycle timescale and the KERs observed can be used to deduce the bond length and bond angles for the X and A states at the time of breakup, separately. The extracted bond length and bond angles have been shown to be in good agreement with those obtained from their modeling.

The idea of extracting the geometry of a polyatomic molecule using the Coulomb explosion imaging (CEI) technique has been around since the last decades and many experiments have been carried out. For a polyatomic molecule, usually only qualitative information can be derived from CEI, for example, for chiral molecules or for isomers. To retrieve the bond lengths or bond angles from CEI is not easy as the details of the excitation-ionization mechanisms and the breakup pathways depend on the molecular structure, including the initial state and all the intermediate states. Using D₂O as the target, with linearly and circularly polarized lasers and for several intensities, the authors was able to "isolate" possible pathways leading to the final momentum correlation observed. In this respect, this MS has made significant advance in imaging molecular structures using the CEI if the analysis and simulations steps are justified.

To obtain their simulation results, the authors have made a lot of assumptions in the model calculations. Since the modelling plays a significant role in deriving their conclusions, the details should be well documented. A reviewer needs to go over all of these models and examine them in order to support this MS for publication. Granted that such details may not be needed in the main text for the general readers, but they should be provided in the Supplemental materials. In the submitted Supplementary Information, Section III for Simulation Details is too brief. Details of the simulation and justification of the assumptions should be provided in order to support the validity presented in this MS. This is not the case. Clearly, the authors have done the simulation and they should be able to provide such information.

At this stage, I am not prepared to give this MS a positive recommendation. If the authors can document that the simulation details are on firm ground, then this work would be a great step forward for showing that it is possible to extract dynamic molecular imaging using the CEI.

In the following I list a few questions that the authors need to clarify. If the authors have the opportunity to revise the MS, they should be considered.

I will use 2/8 to mean page 2 of the main text and 3'/11 to mean page 3 of the Supplemental text, for example.

Page 2/8. The paragraph after the caption of Fig. 1 is too dense and difficult to follow at the first reading. I would suggest that you first discuss the difference among the three regions, then focus on region III. After that, then you talk about region I and II. In the present version, you take the readers to I and II, and then III and then back to I and II.

Page 2'/11. This figure was presented but nothing has been said on page 2/8. A sentence or two should be given to make your points.

2nd paragraph from the end of page 2/8. Figs. 2(c) and 2(d) should be Figs. 1(c) and 1(d). Fig. 1(e)

in the figure should be 1(d).

Page 2/8. Line -12 from bottom. Please clarify what is "filtered" distributions. It would be useful to give a short section in the Supplement to demonstrate this procedure since data "manipulation" will affect the final result.

Page 2/8, second paragraph from the end. This paragraph is very confusing. Besides the previous comment, it is not possible to understand what has been done in the treatment of the data. In Fig. 1, there is a panel for (e), but none for (d). A new section should be added to the Supplement to document the data "filtering". Make sure all the figures are numbered correctly in the captions as well as in the text.

Last paragraph on page 2/8 and Fig. 2(a). The sketch in Fig. 2(a) is very confusing. There is no way to understand that the black arrow indicates double ionization from the cation to trication via collision with an electron wave packet. It is better to draw the arrow from the cation to trication and explain the "mechanism" in the caption for Fig. 2a.

Additional problems in Fig. 2(a). (1) Nothing has been said about the curves for 105 and 150, presumably they are the angle between the two D ions. Have to say why these two angles were chosen. (2) The spectroscopic notations for the trication used in Fig. 2(a) in Table 1 are different from those used in Table 2 and Fig. 2(a). Either correct them or clarify.

The examples above only focus on page 2/8.

In reading the manuscript, I have tried to understand how the simulations have been done. The descriptions on page 6/8 and 4/11 are absolutely inadequate. These two parts have to be rewritten. I should mention that Fig. 2b in the main text and Fig. 4 and 5 in the Supplement are not very illuminating especially since they would depend on the orientation of the molecule. What happens at a different orientation?

In summary, if this MS is to be further considered for Nature Communications, it has to be seriously revised to provide the simulation details since details used to interpret the experimental observed data.

Reviewer #2:

Remarks to the Author:

The current manuscript shows the geometry reconstruction of water in the ionised state in the strong field using Coulomb explosion data. The authors illuminate heavy water from a supersonic source with 40 fs pulses in the near-infrared, achieving intensities in the range of 10^{14} W/cm². Single-charged O and D fragments arising from tri-cationic states are detected. The differences between linear and circular polarisation are due to double ionisation induced by electron recollision. A normalisation procedure eliminates the signal contributions from sequential - not rebound-induced - processes. A fitting procedure in the KER theta plane is used to derive the bending angle and the KER in the multiple ionisation processes. A simulation shown in Fig. 1 e is used to confirm the correct assignment of the different ionisation channels. The KER and the simulated PES allow the assignment of the measured molecular angles to the cationic states X and A.

The authors show that a pure Coulomb model to derive the intranuclear separations fails and leads to inconsistent results for the initial A and X states. Therefore, the authors use a full potential energy surface of the high-level tricationic state to derive the correct geometries.

I congratulate the authors for being able to resolve the geometry of the molecule in the changing laser field. Some technical details still need to be explained in more detail to convince me of the technical soundness of this work. These points are explained below.

Most importantly, I feel that this paper contains too many overstated attributes in terms of an editorial decision. The introduction contains statements about photoexcited molecules and the importance of non-adiabatic dynamics. None of this is addressed in the present manuscript. Furthermore, the introduction contains bold statements about what is not currently covered by other methods. When it says: "However, capturing excited state-resolved transient molecular structures with current ultrafast imaging techniques remains challenging because their accuracy or sensitivity is limited.", one has to put this in the context of the sensitivity required. Another statement about conventional (X-ray and electron) diffraction and their insensitivity to hydrogen is simply wrong. Furthermore, these techniques allow for time-resolved geometry deduction, which is lacking in the current method.

The main results of the work are that the A-state activates molecular bending, the X-state activates stretching. This is already evident from standard photoelectron spectroscopy. In addition, the authors

show that they perform a high-level ab initio theory, another technique for fully resolving water dynamics in field ionised potentials.

I recommend that this work be published in a more appropriate journal specialising in strong-field physics.

As for the simulations, I wonder how the authors deal with the relative angle between the molecule and the field. The MO-ADK theory takes this into account, and the ionisation rate depends on the angle. Also, the angular distribution could change as a function of time in a 40 fs laser field.

Furthermore, I wonder how the spatial intensity profile is included in the simulations. A 2d Gaussian most likely describes the focus, and different intensities in the profile will affect the observation, which is not described in the manuscript.

We thank the reviewers for their thoughtful comments, which have motivated a major revision and substantial improvements of our manuscripts. Particularly, we have performed new experiments of Coulomb explosion imaging (CEI) results of D₂O initiated by electron impact triple ionisation. These results enabled the capturing of the ultrafast nuclei relaxation induced by double inner-shell ionisation and retrieval of the accurate geometry of neutral D₂O experimentally, which are essential but not possible in the CEI induced by femtosecond laser pulse. Combined those results together, we obtained the direct imaging of the electronic excited state-resolved transient structures for cation and dication, and revealed the ultrafast structural evolution triggered by both the valence and inner-shell ionisation within 10 femtoseconds incorporating with the theoretical simulations. Moreover, we clearly documented all the details about our simulations in the supplementary Information. The present version of the manuscript can well address all the comments from both reviewers.

In the following, we provide a point-to-point reply to the comments of the reviewers. Their comments are reproduced in black, our replies in blue and the changes made to the manuscript in green.

Reviewer #1 (Remarks to the Author):

Report on NCOMMS-21-31786E

“Directly imaging excited states-resolved coherent nuclear motions of water with picometer-femtosecond precision” by Zhenzhen Wang et al

In this manuscript, the authors used a laser of 800 nm, 40 fs, 1 kHz, intensity in the range of $3\text{-}6 \times 10^4 \text{ W/cm}^2$ to multiply ionize D₂O molecules and measured the three-dimensional momentum vectors of the ions, D⁺, D⁺ and O⁺. By analyzing the correlation of the total kinetic energy release (KER) and the angle (Θ) between the two D⁺ ions, they obtained the KER- Θ plots, for linearly and circularly polarized lights. They were able to distinguish three KER- Θ regions (called I, II and III), depending on the high/low KER's and small/large angles. They focused on regions I and II for the D⁺/D⁺/O⁺ breakup of D₂O³⁺ ions, distinguishing that region I is due to symmetric breaking of the two D⁺ ions, while region II is due to the asymmetric breakup. In their model, the D₂O neutral is first tunnel ionized by the laser, creating D₂O⁺ in the ground state X and the first excited state A. The D₂O⁺ is subsequently further doubly ionized by laser-induced electron rescattering to form dissociative D₂O³⁺ ions. Based on this idea and additional assumptions, they stated that the experimental results could be attributed to the breakup of D₂O³⁺ ions at sub-cycle timescale and the KERs observed can be used to deduce the bond length and bond angles for the X and A states at the time of breakup, separately. The extracted bond length and bond angles have been shown to be in good agreement with those obtained from their modeling.

The idea of extracting the geometry of a polyatomic molecule using the Coulomb explosion imaging (CEI) technique has been around since the last decades and many experiments have been carried out. **For a polyatomic molecule, usually only**

qualitative information can be derived from CEI, for example, for chiral molecules or for isomers. To retrieve the bond lengths or bond angles from CEI is not easy as the details of the excitation-ionization mechanisms and the breakup pathways depend on the molecular structure, including the initial state and all the intermediate states. Using D₂O as the target, with linearly and circularly polarized lasers and for several intensities, the authors was able to “isolate” possible pathways leading to the final momentum correlation observed. **In this respect, this MS has made significant advance in imaging molecular structures using the CEI if the analysis and simulations steps are justified.**

To obtain their simulation results, the authors have made a lot of assumptions in the model calculations. Since the modelling plays a significant role in deriving their conclusions, the details should be well documented. A reviewer needs to go over all of these models and examine them in order to support this MS for publication. Granted that such details may not be needed in the main text for the general readers, but they should be provided in the Supplemental materials. In the submitted Supplementary Information, Section III for Simulation Details is too brief. Details of the simulation and justification of the assumptions should be provided in order to support the validity presented in this MS. This is not the case. Clearly, the authors have done the simulation and they should be able to provide such information.

At this stage, I am not prepared to give this MS a positive recommendation. **If the authors can document that the simulation details are on firm ground, then this work would be a great step forward for showing that it is possible to extract dynamic molecular imaging using the CEI.**

Reply: We thank the reviewer for the accurate and comprehensive summary. In the new version of the manuscript, we provided all the details and listed all the involved assumptions of our simulation approach in the Supplemental Information. Moreover, we implemented new CEI measurement of D₂O driven by 200 eV pulsed electron beam. The autoionization state of dication can be populated by the removal of two inner-shell electrons upon electron impact, and the following ultrafast nuclear motion has been retrieved with the help of simulations. Additionally, three valence electrons of D₂O can be directly kicked out by electron impact, which can fully freeze all the nuclear motion during the triple ionisation. The three-body CE results from these vertical triple ionisation channels can be well reproduced by the simulations, which can further verify our simulation approach. Combining those results from TERCE and electron impact, we have visualized the ultrafast nuclei dynamics of ionized water triggered by both valence and inner-shell ionisation.

In the following I list a few questions that the authors need to clarify. If the authors have the opportunity to revise the MS, they should be considered.

I will use 2/8 to mean page 2 of the main text and 3'/11 to mean page 3 of the Supplemental text, for example.

Page 2/8. The paragraph after the caption of Fig. 1 is too dense and difficult to follow at the first reading. I would suggest that you first discuss the difference among the three regions, then focus on region III. After that, then you talk about region I and II. In the present version, you take the readers to I and II, and then III and then back to I and II.

Reply: We fully agree to the reviewer, and changed this part accordingly. In the current version of the manuscript, the region I with weak polarization dependence is firstly discussed, and then we focused on the region II and III with strong polarization dependence.

Changes: in the Results section, the differences between region I and II are compared at the first paragraph in more details, "The region I (centred at ~ 20 eV)".

Page 2/11. This figure was presented but nothing has been said on page 2/8. A sentence or two should be given to make your points.

Reply: we added a new paragraph in Results section to describe this figure.

Changes: "To further reveal the dissociation dynamics, we can further retrieve the transient structure before the TERCE".

2nd paragraph from the end of page 2/8. Figs. 2(c) and 2(d) should be Figs. 1(c) and 1(d). Fig. 1(e) in the figure should be 1(d).

Reply: Thanks for the reviewer's careful review. We revised them accordingly.

Page 2/8. Line -12 from bottom. Please clarify what is "filtered" distributions. It would be useful to give a short section in the Supplement to demonstrate this procedure since data "manipulation" will affect the final result. Page 2/8, second paragraph from the end. This paragraph is very confusing. Besides the previous comment, it is not possible to understand what has been done in the treatment of the data. In Fig. 1, there is a panel for (e), but none for (d). A new section should be added to the Supplement to document the data "filtering". Make sure all the figures are numbered correctly in the captions as well as in the text.

Reply: Thanks for the reviewer's valuable suggestions. In the revised manuscript, we selected events from three-body electron recollision-assisted Coulomb explosion (TERCE) (region II and III) based on the criteria of energy correlations between fragments, and discard the intensity filtering method used in the first version. The current data analysis method is widely used in the ion-ion coincidence momentum measurement, such as [P. Corkum group, PRL, 91, 093002, 2003; J. Ullrich group, PRL, 97,193001, 2006].

In Fig. 1(a) of the main text, the counts from the strong-field enhanced ionisation (region I) are around 100 times higher than that from TERCE (region II and III). To extract the peak

positions of different channels of TERCE in (θ_{cm} -KER) distributions through the fit, we need to isolate the events in region II and III from the region I (see Fig. R1(a)).

The symmetric three-body CE process is dominant in region II (see Supplementary Fig. 1(a)). For D_2O , two D^+ carry most of the energy from CE, and the energy of O^+ is much smaller. The one-dimensional distributions of the kinetic energy difference between D^+ and O^+ for all the events are shown in Fig. R1(b) for linearly and circularly polarisations. A significant yield hump in the high energy tail can be seen for linear polarization, and in contrast, a prominent drop of yield in this energy range appears for circular polarisation, and the two curves cross at 12 eV. Thus the events from TERCE can be selected by setting the condition of the energy differences of D^+ and O^+ ($\text{KE}_{\text{D}^+} - \text{KE}_{\text{O}^+}$) being larger than 12 eV. Moreover, the energy difference between two D^+ ($|\text{KE}_{\text{D}1^+} - \text{KE}_{\text{D}2^+}|$) in region II is mostly less than 6 eV, as shown by the dashed line in Fig. R1 (c) and Fig. R2 (a). By selecting the events with $(\text{KE}_{\text{D}^+} - \text{KE}_{\text{O}^+})$ larger than 12 eV and $(\text{KE}_{\text{D}1^+} - \text{KE}_{\text{D}2^+})$ smaller than 6 eV, we present the (θ_{cm} -KER) distribution in Fig. R1 (d). In Fig. R1 (d), the remained counts from region I mainly located at the corner with low KER and larger θ_{cm} , which can allow us to do the two-dimensional fit and isolate the different components from the region II.

For asymmetric CE in region III, the energy difference between two D^+ is significantly larger than symmetric CE, as shown in the Fig. R2 (a). The energy correlation map clearly separates the region III and other regions; thus we used this feature to select events from region III. The events with $(|\text{KE}_{\text{D}1^+} - \text{KE}_{\text{D}2^+}|) > 6$ eV are selected and their distributions are shown in Fig. R2 (c). The value of 6 eV (indicated by the pink dashed line in Fig. R2 (a) and (b)) is chosen here simply because nearly no counts can be found beyond 6 eV in the case of circular polarization, as can be seen in Fig. R2 (b).

We summed up the selected events for region II and III from TERCE together and the total distributions are shown in Fig. R3 (a). Based on this distribution, we can obtain the peak positions of the dominant components from TERCE with two-dimensional fitting procedure. In Fig. R3 (b), we present the (θ_{cm} -KER) distribution obtained in the previous version, which is the intensity ratio between the distribution of linear and circular polarization. In Fig. R3 (c), we present the (θ_{cm} -KER) distribution obtained by directly subtracting the distribution of circular polarization from that of the linear polarization after normalization. Comparing those three distributions, they are overall similar for higher KER region (>27 eV), however, only panel (a) contains the distributions around 26 eV, since the data analysis methods used in (b) and (c) are severely affected by the overlapping region between region I and II.

Based on the characteristic energy correlation of three fragments from TERCE, we can isolate the events from symmetric CE in region II and asymmetric CE in region III. In this way, we can obtain the (θ_{cm} -KER) peak positions of dominant components in region II and region III through the fit.

Fig. R1. (a) presents the measured raw (θ_{cm} -KER) distributions of all the events from three-body CE triggered by the linearly polarised laser pulse. (b) shows the one-dimensional distributions of $(KE_{D1^+} - KE_{D2^+})$ for all the event in the case of linearly (LP) and circularly polarised (CP) laser pulses. Two curves cross at 12 eV as indicated by the dashed line. (c) One dimensional kinetic energy differences between two D^+ are present for region I and region II in the case linearly polarised laser pulse. (d) presents the (θ_{cm} -KER) distributions after applying the conditions of $(KE_{D^+} - KE_{O^+}) > 12$ eV and $(|KE_{D1^+} - KE_{D2^+}|) < 6$ eV.

Fig. R2. (a) and (b) show the energy correlation between two D^+ in the case of linearly and circularly polarised laser pulse, and the dashed lines stand for $(|KE_{D1^+} - KE_{D2^+}|) = 6$ eV. (c) shows (θ_{cm} -KER) distribution (region III) after applying the condition of $(|KE_{D1^+} - KE_{D2^+}|) > 6$ eV.

Fig. R3. (a) The $(\theta_{\text{cm}}\text{-KER})$ distribution of selected events by applying conditions of energy correlation between fragments (same as Fig. 1(c) in the main text). (b) The $(\theta_{\text{cm}}\text{-KER})$ distribution is derived by obtaining the intensity ratio between the raw $(\theta_{\text{cm}}\text{-KER})$ distribution of LP and that of CP, i.e., $(\text{Intensity})_{\text{LP}}/(\text{Intensity})_{\text{CP}}$. (c) presents the results after subtracting the $(\theta_{\text{cm}}\text{-KER})$ distribution of CP from that of LP, i.e., $(\text{Intensity})_{\text{LP}} - (\text{Intensity})_{\text{CP}}$ after normalization. The vertical dashed line stands for the position of 26 eV.

Changes: In the main text, Fig.1 (c) is added to present the $(\theta_{\text{cm}}\text{-KER})$ distributions which satisfy the applied conditions, and Fig.1(d) showed that the $(\theta_{\text{cm}}\text{-KER})$ distributions applying the same conditions in the case of circular polarization. Fig. 1(e) presents the simulations and Fig.1(f) presents the two-dimensional fit of Fig. 1(c). In the Supplementary files, we added Fig. 1 and Fig. 2 in section 1 to explain the details of our data analysis procedure.

Last paragraph on page 2/8 and Fig. 2(a). The sketch in Fig. 2(a) is very confusing. There is no way to understand that the black arrow indicates double ionization from the cation to trication via collision with an electron wave packet. It is better to draw the arrow from the cation to trication and explain the “mechanism” in the caption for Fig. 2a.

Reply: we redrew the sketch in Fig. 2(a) and Fig.3(c), and explain the mechanism in the captions.

Changes: In Fig. 2(a), we draw the arrow from cation to trication and labeled it as nonsequential double ionisation from cation (NSDI). In Fig. 3(c), we draw the arrow from ground state to the C state of cation and labeled it as electron recollision induced excitation (RIE).

Additional problems in Fig. 2(a). (1) Nothing has been said about the curves for 105 and 150, presumably they are the angle between the two D ions. Have to say why these two angles were chosen. (2) The spectroscopic notations for the trication used in Fig. 2(a) in Table 1 are different from those used in Table 2 and Fig. 2(a). Either correct them or clarify.

The examples above only focus on page 2/8.

Reply: (1) 105° is the equilibrium bond angle of D_2O , for the X state, the calculated most probable double ionisation occurs around this bond angle. 150° is the bond angle of A state of cation at the most probable double ionisation instant. (2) We changed the spectroscopic notations of the trication in Table 1.

Changes: we added the explanation in the end of the second paragraph on page 4/13, and corrected the spectroscopic notations in Table 1.

In reading the manuscript, I have tried to understand how the simulations have been done. The descriptions on page 6/8 and 4/11 are absolutely inadequate. These two parts have to be rewritten.

Reply: We have modified the details of the simulation method in the main text and the Supplementary Information.

Changes: in the second and third paragraphs on page 4/13 of the main text, we rewrote the description of the simulation procedure. In the Supplementary Information, we first describe how we did the simulation in section III, and listed all the assumptions used in the simulation in section IV, finally, we present the simulated results (Supplementary Figure 4 to Figure 10) in a step-by-step manner.

I should mention that Fig. 2b in the main text and Fig. 4 and 5 in the Supplement are not very illuminating especially since they would depend on the orientation of the molecule. What happens at a different orientation?

Reply: according to the reviewer's comment, we have performed the simulation of the time-dependent evolution of correlated vibrational wave packet $\Psi(r_1, r_2, \theta)$ and rotational wave packet $\Phi(\alpha, \beta, \gamma)$ in the revised manuscript. Due to the limitation of computing resources, we cannot directly perform the time-dependent evolution of six-dimensional vibrational-rotational wave packets simultaneously. In order to simplify the calculation, we use two approximations to separate the evolution of molecular vibration and rotational wave packets. i) The ionisations induced by laser and electron collision both satisfy the Franck-Condon principle, namely the nuclear vibrational wave packets $\Psi(r_1, r_2, \theta)$ are kept frozen before and after the ionisation; ii) The laser field can be considered as a perturbation interaction to the orientational distribution $\Phi(\alpha, \beta, \gamma)$, but the laser field cannot affect the evolution of nuclear vibrational wave packet $\Psi(r_1, r_2, \theta)$ along the potential energy surfaces. Based on those two assumptions, we can assume that the evolution of vibrational wave packets $\Psi(r_1, r_2, \theta)$ only depends on the time intervals between single and double ionisation triggered by the laser pulse. Moreover, molecular orientational distribution $\Phi(\alpha, \beta, \gamma)$ will obviously depend on the evolution of vibrational wave packet $\Psi(r_1, r_2, \theta)$ since the geometry, rotational inertia and dipole polarizability of water molecule will change obviously with the evolution of the vibration wave packet. More importantly, the molecular ionisation potential and the orbital wave function of ionised electrons will also change significantly, thus affecting

the angular probability of multiple ionisation.

In our simulation, we first simulate the evolution of $\Psi(r_1, r_2, \theta)$ from D₂O to D₂O³⁺ with different ionisation interval values firstly. The results have been shown in Fig.2 in the main text and Figure 4 and Figure 5 in Supplementary Information. Here, we can obtain the time-dependent average geometry, rotational inertia, dipole polarizability, ionisation potential and orbital wave function of ionised electrons. Based on these factors, we can calculate the ionisation probabilities for the single and double ionisation induced by intense laser pulses and the third ionisation induced by the electron re-collision for all possible triple ionisation time combinations. The partial results have been shown in Figure 7 of Supplementary Information. Then, we simulate the evolution of rotational wave packet $\Phi(\alpha, \beta, \gamma)$ using the time-dependent wave packet method, where we consider the variation of molecular orientation caused by the laser and the correlation between the vibrational wave packets $\Psi(r_1, r_2, \theta)$ and $\Phi(\alpha, \beta, \gamma)$. The partial results have been shown in Figure 8 of Supplementary Information. Finally, we can obtain the weighting factor $W(T)$ as a function of time intervals between two successive laser ionisation according to Equation (4) and (5) in Supplementary Information. The results are shown in Figure 9 of Supplementary Information. Note that the influences of different molecular orientations on the final results have been included in $W(T)$ by integrating the three-dimensional ionisation probability.

Changes: we updated the simulated final results by considering the orientation effect in Fig. 1 (e) in the main text. We described the simulation details in the section IV and presented detailed results in Figure 6 to Figure 10 in the supplementary information.

In summary, if this MS is to be further considered for Nature Communications, it has to be seriously revised to provide the simulation details since details used to interpret the experimental observed data.

Reply: According to the reviewer's suggestions, we provided all the assumptions and details of simulation, and modified the manuscript to improve the readability. In the new version of manuscript, we obtained electronic state-resolved nuclei dynamics information of heavy water with CEI based on the electron impact triple ionisation. Relative to the rich nuclear dynamics in the cation induced by the femtosecond laser pulse, the electron impact ionisation can directly kick out three valence electrons, or kick out two inner-shell electrons followed by Auger decay. The vertical triple ionisation in attosecond regime induced by electron impact provide a benchmark to verify our simulation approach. The good agreement between measurement and simulation for those three-body CE channels after vertical triple ionisation confirms the validity of our simulations and interpretations. Based on the similar analysis method as that used in TERCE, we revealed nuclei relaxation dynamics along an autoionization state of water during Auger

decay. In summary, we revealed the excited state-resolved nuclei dynamics initiated by the ionisation of both inner-shell and valence level electrons with few-femtosecond and picometer spatiotemporal resolution.

Reviewer #2 (Remarks to the Author):

The current manuscript shows the geometry reconstruction of water in the ionised state in the strong field using Coulomb explosion data. The authors illuminate heavy water from a supersonic source with 40 fs pulses in the near-infrared, achieving intensities in the range of 10^{14} W/cm². Single-charged O and D fragments arising from tri-cationic states are detected. The differences between linear and circular polarisation are due to double ionisation induced by electron recollision. A normalisation procedure eliminates the signal contributions from sequential - not rebound-induced - processes.

A fitting procedure in the KER theta plane is used to derive the bending angle and the KER in the multiple ionisation processes. A simulation shown in Fig. 1 e is used to confirm the correct assignment of the different ionisation channels. The KER and the simulated PES allow the assignment of the measured molecular angles to the cationic states X and A.

The authors show that a pure Coulomb model to derive the intranuclear separations fails and leads to inconsistent results for the initial A and X states. Therefore, the authors use a full potential energy surface of the high-level tricationic state to derive the correct geometries.

I congratulate the authors for being able to resolve the geometry of the molecule in the changing laser field. Some technical details still need to be explained in more detail to convince me of the technical soundness of this work. These points are explained below.

Reply: We thank the reviewer for the accurate summary, in the previous version of manuscript, we present the direct imaging of transient structures for three electronic states of water cation with Coulomb explosion imaging technique. By developing three-body electron recollision-assisted explosion imaging (TERCE) technique and a molecular dynamics simulation, the transient structure retrieval based on CEI can reach to a quantitative level. In the revised version of manuscript, we performed new CEI experiments of heavy water induced by 200 eV electron impact ionisation. With the same observable of three-body CE and molecular dynamics simulation as TERCE, we captured the transient structure of autoionization state of dication populated by the removal of two inner-shell electrons. Combining the CEI studies induced by the femtosecond laser pulse and electron beam, the ultrafast structural evolutions triggered by inner-shell and valence level ionisation can be revealed, including the dynamics along the ground state and excited state of cation, dication and trication.

Most importantly, I feel that this paper contains too many overstated attributes in terms of an editorial decision. The introduction contains statements about photoexcited molecules and the importance of non-adiabatic dynamics. None of this is addressed in the present manuscript. Furthermore, the introduction contains bold statements about what is not

currently covered by other methods. When it says: "However, capturing excited state-resolved transient molecular structures with current ultrafast imaging techniques remains challenging because their accuracy or sensitivity is limited.", one has to put this in the context of the sensitivity required. Another statement about conventional (X-ray and electron) diffraction and their insensitivity to hydrogen is simply wrong.

Reply: Thanks for the reviewer's valuable comments. We have modified the introduction section accordingly. The motivation of the current work is to reveal the electronic state-resolved ultrafast molecular dynamics triggered by ionisation of gaseous water with few-femtosecond and few-picometer spatiotemporal resolution based on the Coulomb explosion imaging (CEI) technique.

Changes: we omitted all the arguments mentioned by the reviewer, and rewrote the whole introduction part accordingly in the first two paragraphs in the main text.

Furthermore, these techniques allow for time-resolved geometry deduction, which is lacking in the current method.

Reply: The pump-probe technique allows the time-resolved structural evolution imaging of molecular dynamics, where the femtosecond laser pulse can be used as a pump and the ultrafast electron or X-ray diffraction imaging can be used as a probe (*Science*, 361, 64 (2018)). With these methods, the temporal resolution can reach to several tens of femtoseconds for time-resolved geometry deduction (*PRL*, 124, 134803 (2020)). To push the temporal resolution to the few femtoseconds region, an intracycle pump-probe scheme based on the three-step model in strong field ionisation is proposed by Niikura et al. (*Nature*, 417, 917 (2002)), where the strong field tunnelling ionisation (pump) can initiate a nuclei wave-packet motion along the excited state of D_2^+ , and the transient position of the nuclei is captured by electron recollision-induced double ionisation and Coulomb explosion (probe). The time delay between ionisation and recollision is approximately three quarters of one optical cycle, which can be varied by changing the wavelength of the driving laser pulse. This intracycle pump-probe scheme is known as the 'molecule clock'. In the current TERCE approach, a similar intracycle pump-probe scheme can be achieved, i.e., the tunnelling ionisation of cation triggered the nuclear wave-packet evolution along the ground state of dication, and the electron recollision-induced triple ionisation and CE can be used as a probe. The time delay between the pump and probe is approximately 2 fs for our 800 nm laser pulses. Combined with the simulations, we reveal that the R_{OD} can extend 9 pm within a time delay of 2 fs.

Moreover, recent studies in the T. Jahnke group have proposed an intrapulse pump-probe Coulomb explosion imaging technique with an intense X-Ray Free-Electron laser [*PRX*, 11, 041044, 2021]. In their approach, the single-photon double ionisation at time t_1 serves as a pump, and at later time t_2 , another photon is absorbed and initiated CE, which serves as a probe, and the time delay closely relates to the pulse width of the laser pulse. A similar scheme can be used in our approach. The tunnelling ionisation from neutral molecule at t_1 can serve as a pump, and the electron-recollision induced double ionisation from the cation to trication (within 2 fs) at a later time t_2 can serve as a probe.

The different time delays will lead to different (θ_{cm} -KER) distributions of fragments after Coulomb explosion. The time delay in this approach can be varied by changing the pulse width of the driving laser pulse. In Fig. R3, we simulated (θ_{cm} -KER) distributions for the pulse widths of 5 fs, 10 fs, 15 fs and 20 fs. We can find that by increasing the time delay (promotional to pulse width), the KER decreases and the θ_{cm} expands to a larger angle originating from the time-dependent evolution of the nuclei wave-packet along the cation. In the new version of the manuscript, we added the (θ_{cm} -KER) distribution around time zero, i.e., the direct triple ionisation of neutral ground state induced by the electron impact, and these results can be quantitatively reproduced by our simulations.

In summary, in the revised manuscript, we obtained the accurate imaging of the ground state of the neutral structure (structure at time zero), and show the time-resolved nuclear wave-packet evolution for cation and dication by combining the retrieval of transient structures and theoretical simulations. Based on the intracycle pump-probe scheme and intrapulse pump-probe scheme, the structural evolution along three electronic states of cation and ground state of dication have been revealed in the time scale of a few femtoseconds with ultrafast Coulomb explosion imaging.

Fig. R4. Simulated (θ_{cm} -KER) distributions of region II driven by the laser pulse with the pulse width of 5 fs, 10 fs, 15 fs and 20 fs.

Changes: in the main text, Fig. 4 is added to present the three-body CE results of heavy water induced by the electron impact, and the vertical triple ionisation channels provide the benchmark of (θ_{cm} -KER) distribution at time zero. The time-resolved nuclei dynamics for different ionic states obtained in the present manuscript are summarized in Fig. 5.

The main results of the work are that the A-state activates molecular bending, the X-state activates stretching. This is already evident from standard photoelectron spectroscopy. In

addition, the authors show that they perform a high-level ab initio theory, another technique for fully resolving water dynamics in field ionised potentials.

Reply: we agree with the reviewer that the vibrational modes of X and A states of water can be revealed by the photoelectron spectroscopy with the support of the high-level ab initio calculations. In this manuscript, we propose an alternative way to reveal the ultrafast nuclear wave-packet dynamics along the different electronic states by direct imaging their transient structures with Coulomb explosion imaging technique. By developing TERCE and molecular dynamics simulation method, we can achieve the few-picometers and few-femtoseconds accuracy for the structural retrieval of ionic heavy water, and thus can directly present the time-resolved nuclear motions along different electronic states for both cation and dication. At the instant of 5 fs after the initial population of X state, the R_{OD} extends 7 pm relative to the equilibrium bond length; and after another 2 fs evolution along the ground state of dication, R_{OD} further extends another 4 pm. At the instant of 8 fs after the initial population of A state, the R_{OD} extends 10 pm relative to the equilibrium bond length, and the bond angle increases from 97° to 146° ; after another 2 fs evolution along the ground state of dication, the R_{OD} further extends 9 pm. Those ultrafast dynamics has not been revealed by the electron spectroscopy. More importantly, TERCE approach can explore the unknown complex ultrafast nuclei dynamics. We revealed the large amplitude bending and asymmetric stretching motions along higher excited state (C state), which leads to that the two D atoms locate at the same side of O atom. Such a structure deformation has not been explored by the electron spectroscopy. Finally, with CE imaging technique, we showed the ultrafast motion of nuclei along autoionization state of dication followed by Auger decay, which provide new insights in the ultrafast nuclei relaxation of water after irradiation of high energy electrons or photons.

Changes: Fig. 5 in the main text is added in the new version of the manuscript to summary the time-resolved structural evolution for different electronic states of D_2O .

I recommend that this work be published in a more appropriate journal specialising in strong-field physics.

We thank the reviewer for those detailed comments and for the opportunity to declare the breadth and novelty of our work in terms of the scientific insights.

Capturing the accurate transient structure of a polyatomic molecule is a critical step towards to the direct visualization of ultrafast chemical reactions in real time, which attracts considerable broad interests in ultrafast physics and femtochemistry. Moreover, the water molecule is one of the most important targets, and studying its structure and rich nuclei dynamics induced by the ionisation are critical to understand water-related physics and chemistry, such as the formation of H_3O^+ and OH^* . In the new version of the manuscript,

1) we achieve the structural retrieval of different electronic states for cation and dication of heavy water with CEI in a quantitative level, and obtained the accurate neutral

geometry of D₂O with a direct ultrafast structural imaging technique.

2) Combined the with CEI driven by the intense laser pulse and electron impact with the support of molecular dynamics simulations, we reveal the rich nuclei dynamics after ionisation of valence and inner-shell electrons, showing the large amplitude symmetric and asymmetric motions for A and C states of cation and the ultrafast nuclei relaxation before Auger ionisation for the autoionization state of dication.

3) Our results also suggest that the time scale of Auger decay here is 3.2 fs.

4) For the three-body CEI triggered by both strong field ionisation and electron impact ionisation, the doublet and quartet state of trication both can have important contributions, which have been confirmed in this work for the first time.

5) Our simulation method can well reproduce the experimental results from strong field ionisation and electron impact ionisation, and thus show a potential for the general application to explore the ultrafast nuclei dynamics.

6) The proposed TERCE can achieve the electronic state-resolved transient structure retrieval with few-picometer and few-femtosecond accuracy.

Changes: new three-body CEI experiment of D₂O with electron impact ionisation is added and the results are shown in Fig. 4 in the main text. We summarized the time-resolved structural evolution in Fig. 5 in the main text. We also changed the title, introduction, and conclusion accordingly.

As for the simulations, I wonder how the authors deal with the relative angle between the molecule and the field. The MO-ADK theory takes this into account, and the ionisation rate depends on the angle. Also, the angular distribution could change as a function of time in a 40 fs laser field.

Reply: Thanks for reviewer's valuable comments. Yes, we used MO-ADK theory to consider the angle-dependent tunnelling ionisation yield in the femtosecond laser pulse. Moreover, we improved our simulation approach by including the effect of molecular orientation in our results. We have performed the simulation of the time-dependent evolution of correlated vibrational wave packet $\Psi(r_1, r_2, \theta)$ and rotational wave packet $\Phi(\alpha, \beta, \gamma)$. Due to the limitation of computing resources, we cannot directly perform the time-dependent evolution of six-dimensional vibrational-rotational wave packets simultaneously. In order to simplify the calculation, we use two approximations to separate the evolution of molecular vibration and rotational wave packets. i) The ionisations induced by laser and electron impact both satisfy the Franck-Condon principle, namely the nuclear vibrational wave packets $\Psi(r_1, r_2, \theta)$ are frozen before and after the ionisation; ii) The laser field can be considered as a perturbation interaction to the orientational distribution $\Phi(\alpha, \beta, \gamma)$, but the laser field cannot affect the evolution of nuclear vibrational wave packet $\Psi(r_1, r_2, \theta)$ along the potential energy

surfaces. Based on those two assumptions, we can assume that the evolution of vibrational wave packets $\Psi(r_1, r_2, \theta)$ only depends on the time intervals between single and double ionisation triggered by the laser pulse. Moreover, molecular orientational distribution $\Phi(\alpha, \beta, \gamma)$ will obviously depend on the evolution of vibrational wave packet $\Psi(r_1, r_2, \theta)$ since the geometry, rotational inertia and dipole polarizability of the water molecule are changing largely with the evolution of the vibration wave packet. More importantly, the molecular ionisation potential and the orbital wave function of ionised electron will also change significantly, thus affecting the angular probability of multiple ionisation.

In our simulation, we first simulate the evolution of $\Psi(r_1, r_2, \theta)$ from D_2O to D_2O^{3+} with different ionisation interval values firstly. The results have been shown in Fig.2 in the main text and Figure 4 and Figure 5 in Supplementary Information. Here, we can obtain the time-dependent average geometry, rotational inertia, dipole polarizability, ionisation potential and orbital wave function of ionized electron. Based on these factors, we can calculate the ionisation probabilities for the single and double ionisation induced by intense laser pulses and the triple ionisation induced by the electron re-collision for all possible triple ionisation time combinations. The partial results have been shown in Figure 7 of Supplementary Information. Then, we simulate the evolution of rotational wave packet $\Phi(\alpha, \beta, \gamma)$ using the time-dependent wave packet method, where the molecular orientation change caused by laser and the correlation between the vibrational wave packets $\Psi(r_1, r_2, \theta)$ and $\Phi(\alpha, \beta, \gamma)$ are fully considered. The partial results have been shown in Figure 8 of Supplementary Information (see Fig. R5). Finally, we can obtain the weighting factor $W(T)$ as a function of time intervals between single and double ionisation according to equation (4) and (5) in Supplementary Information. The results are shown in Figure 9 of Supplementary Information. Thus the influences of different molecular orientations on the final results have been included in $W(T)$ by integrating the three-dimensional ionisation probability. The final results are presented in Fig. 1(e) of the main text.

Fig. R5. Time-dependent angular distribution for different orientation angles (α) of neutral D_2O under the influence of laser field, where α is the angle between the laser field and the

axis perpendicular to the molecular plane.

Changes: Fig. 1(e) in the main text is updated to include the molecular orientation effect in the simulation, and the details of the simulation are added in section III, IV and V of the Supplementary Information.

Furthermore, I wonder how the spatial intensity profile is included in the simulations. A 2D Gaussian most likely describes the focus, and different intensities in the profile will affect the observation, which is not described in the manuscript.

In the present simulations, we did the simulations of region II from TERCE for different laser intensities including 100 TW/cm², 250 TW/cm² and 300 TW/cm², and the results are shown in Fig. R6. It can be seen that those results are quite similar and show weak dependence on peak laser intensity. The (θ_{cm} -KER) distributions from TERCE mainly depends on the relative ionisation probability for different time intervals between single and double ionisation, which is weakly related to the peak laser intensity. Varying the laser intensity can change the total ionisation probability for both single and double ionisation, which has little effect on the relative (θ_{cm} -KER) distributions ionized from X and A state. Thus we didn't include the spatial intensity profile due to the limitation of our computing resources.

Fig. R6. Simulated (θ_{cm} -KER) distributions of the three-body CE channel induced by laser with the intensity of 100 TW/cm² (a), 250 TW/cm² (b) and 300 TW/cm² (c). The distributions are normalized to the maximum value of the distribution.

Changes: Figure 11 is inserted in the Supplementary Information to present the simulated results for different peak laser intensities.

REVIEWER COMMENTS

Reviewer #1 (Remarks to the Author):

Please see attached file

Reviewer #2 (Remarks to the Author):

I would like to thank the authors for taking into account my comments. I am happy about the corrections that have been performed. The authors went to great lengths and even performed new experiments. I certainly appreciate also the technical details in the supporting information. The introduction is at the right level now, the messages are clearly supported by the graphs and quantitative information is deduced.

Reviewer 1, second report

I have read the revised manuscript and the rebuttal from the authors to my earlier report and the report from reviewer 2. The authors have gone through many details in their simulations. Based on the simulations, they claimed that they are able to follow the dissociation pathways versus time after D₂O molecules are first tunnel ionized by an intense laser. Experimentally, they collected the energy and angle of each of the three single-charge atomic ions, D⁺, D⁺ and O⁺. By carrying out theoretical simulations, they concluded that the different three-body breakup follows the pathways and the temporal dependence as summarized in Fig. 5.

These are very strong statements.

Qualitative interpretation of Coulomb explosion experiments has been around for more than two decades. The simulations reported by the authors are much more complete, but that does not mean their conclusions as given in Fig. 5 are indeed how it happened. In the simulations, there are so many approximations. How these approximations would affect the conclusion from the simulation?

The authors have summarized nine approximations used in their simulations on page 7 of the supplementary document. Many of those approximations have not been justified. Here I address a few serious ones.

1. The driving laser is 40 fs duration, thus coupling of molecular states in the laser field would play a role. This issue has not been addressed. Would the temporal evolution be modified if a shorter pulse (say 20 fs or less) is applied?
2. In the NSDI calculation, the so-called MCTDW model was used to calculate the electron impact ionization probability. This model is based on the distorted-wave approximation; thus, it is not expected to work for electron impact on molecular ions and for low energy electrons. Since the returning electrons have a range of energies, integration over the returning electron energy is required. Has this been done?
3. For NSDI, it is well-known that electron impact excitation could be more important than electron impact ionization. Since the excited electron is readily ionized in later cycles, so impact excitation is also treated as NSDI. This possibility was not addressed and it would change the conclusion of the simulation.

There are other issues in their “nine approximations” and others addressed above. Their simulation model has been much more detailed than what is available so far for retrieving dynamics from CEI data. Granted that these approximations might not “hurt” the dissociation dynamics extracted from their simulations. On the other hand, to establish the validity of the present simulation model, the best test is to apply the simulation using different laser parameters, to see if different temporal dynamics (similar to Fig. 5) can be found and then to do CEI experiments with those laser parameters. If this is carried out, then the validity of the approximations used in the simulation is established, and the retrieved molecular dynamics versus time can then be more believable.

In summary, despite that the authors have addressed what are the approximations made in their simulation, it is still “dangerous” **not** to worry that their simulation is not good enough. To make the claim that the molecular dynamics extracted from the CEI data from their simulation would require additional tests from additional experiments. To claim a “big” progress on a long-standing problem

requires higher standard. Such a big claim has been made in this manuscript, but more evidence is needed. In its present form, I have difficulty to recommend its publication in Nature Communications.

Comments to the reply to reviewer 1.

Please refer to the page number of the rebuttal file submitted by the authors. Important comments are in **boldface**.

Page 2.

Comments on CEI measurements by 200 eV electrons are irrelevant. Triple ionization in this case can be due to electron impact to autoionization state of dication (followed by Auger) or by direct triple ionization. These processes are different from the triple ionization by D2O by intense laser discussed in the paper. Their answer to this question is misleading and cannot be correct. I also do not see how you can calculate triple ionization and what autoionizing states were included in the simulation. **This answer makes me worry about the “simulations” reported in this MS.**

Page 3 to page 6.

Thanks for explaining the details of how the data are analyzed and for clarifying that Region II and III are about 100 times smaller than region 1.

Bottom of page 6

It would be essential to address what is the energy range of the rescattering electrons. In calculating the ionization rate for NSDI, one has to integrate over the energies of the rescattering electrons. How these are done and how the molecular orientation is included?

Rescattering excitations are often more important than rescattering ionization for NSDI. The excited electron in general will be ionized at subsequent laser cycles. How to justify it does not contribute to the NSDI?

Page 7

The statement, “Due to the limitation of computing resources...”, is not an excuse for not answering my question. **Why rotation is so important in a laser field for time scale of a few femtoseconds? I would expect that at such short time scale, the electronic dynamics is more likely modified by the laser.** Since the authors do not account for or examine laser couplings among the electronic states, it is not clear how to justify treating the “evolution” of each excited electronic states as independent. **This is actually a critical assumption that should be addressed.**

We thank the reviewers for the valuable comments and suggestions, accordingly, we added new experimental results and simulations with different pulse durations in our manuscript. Specifically, we compared the measured and simulated ($\theta_{\text{cm-KER}}$) distributions from three-body electron recollision-assisted Coulomb explosion (TERCE) induced by the laser pulse with the available pulse durations of 80 fs, 40 fs and 30 fs in our laboratory, and the consistency between measurements and simulations confirms the validity of our simulations. We simulated TERCE with pulse durations from 20 fs to 5 fs and found the bending motion in the A state of water cation is frozen in the case of 5 fs, which leads to the shrink of the distribution of θ_{cm} . This simulated result is validated by the electron-impact-induced vertical triple ionization, where the nuclear motion in the cation is completely frozen. As the reviewer pointed out, when the driving laser pulse is shorter than 10 fs, the time-dependent nuclear evolution correlates to the pulse duration, and thus we revised Fig. 5 and the related discussions in the revised manuscript. Moreover, we further justified some approximations in the simulation as suggested by the reviewer and improved our simulations accordingly. By adding additional measurements and simulations, we believe the present version of the manuscript can address all the comments from the reviewer.

In the following, we provide a point-to-point reply to the comments of the reviewer. The comments are reproduced in black, our replies in blue and the changes made to the manuscript in green.

Reviewer #1:

I have read the revised manuscript and the rebuttal from the authors to my earlier report and the report from reviewer 2. The authors have gone through many details in their simulations. Based on the simulations, they claimed that they are able to follow the dissociation pathways versus time after D_2O molecules are first tunnel ionized by an intense laser. Experimentally, they collected the energy and angle of each of the three single-charge atomic ions, D^+ , D^+ and O^+ . By carrying out theoretical simulations, they concluded that the different three-body breakup follows the pathways and the temporal dependence as summarized in Fig. 5.

These are very strong statements.

Reply: In the revised manuscript, we implemented new measurements and simulations with different pulse durations, and further justified the approximations used in the simulation approach. For this TERCE process where the ionization from cation to dication occurs within a time scale of ten fs, the validity of the simulation approach has been further verified. As the referee pointed out, when the pulse duration is shorter than 10 fs, we found that the time-dependent structural evolution of D_2O depends on the pulse durations. Thus we modified Fig. 5 in the main text to summarize more precisely the structural evolutions. We also revised the related sentences in the abstract, discussions and conclusion to make the statements more accurate, preventing any over-interpretations. More evidences and discussions to support the validity of present

simulations are added in the Supplementary information. All the details about those modifications and changes are listed in the following.

Qualitative interpretation of Coulomb explosion experiments has been around for more than two decades. The simulations reported by the authors are much more complete, but that does not mean their conclusions as given in Fig. 5 are indeed how it happened. In the simulations, there are so many approximations. How these approximations would affect the conclusion from the simulation?

The authors have summarized nine approximations used in their simulations on page 7 of the supplementary document. Many of those approximations have not been justified. Here I address a few serious ones.

1. The driving laser is 40 fs duration, thus coupling of molecular states in the laser field would play a role. This issue has not been addressed.

Reply: Thanks for the valuable suggestions. We agree with the reviewer that the coupling of molecular electronic states in the laser field may play a role during the multiple ionization of water. Here, we tried to evaluate the influences of the possible coupling between ground and excited states of cation in the TERCE.

The X and A states of cation can be coupled by absorbing or emitting photons in the laser field via dipole transition. The probability of the dipole transition between two states can be qualitatively estimated by the dipole transition matrix element in the perturbation regime as follows:

,

$$\begin{aligned}
 C_m(t) &= -\frac{i}{\hbar} \int_0^{t_1} e^{i\omega_{mn}t} H'_{mn} dt \\
 &= \frac{1}{2\hbar} \langle \phi_m^0 | (|ED| \cos\theta) | \phi_n^0 \rangle \int_0^{t_1} [e^{i(\omega_{mn} + \omega)t} + e^{-i(\omega_{mn} - \omega)t}] dt \\
 &= \frac{1}{2\hbar} \langle \phi_m^0 | (|ED| \cos\theta) | \phi_n^0 \rangle \left[\frac{e^{i(\omega_{mn} + \omega)t} - 1}{\omega_{mn} + \omega} + \frac{e^{i(\omega_{mn} - \omega)t} - 1}{\omega_{mn} - \omega} \right]
 \end{aligned}$$

The coupling of the two electronic states can be significant if the resonance transition condition can be met and the transition dipole moment between the two states is large. First, we check whether the resonance transition condition can be met for the coupling of X and A states in the present condition. For the dipole transition from X to the A state via absorbing photon, we first calculated the excited energy between X and A states. The X state can be populated by the tunneling ionization, as shown in Fig. 2(b) in the main text, the vibrational wave-packet of X state localized to a small range of bond angle and bond length. The energy difference between X and A states (D_2O^+) nearly stays constant, which is 0.7 eV larger than the photon energy (1.55 eV for 800 nm pulse) (see Fig. R1), thus the resonance condition of dipole transition can not be met, and the probability of transition from X to A state is expected to be small. For the dipole transition from A state to X state via emitting photon, the initial wave-packet of A state is populated by the direct tunneling ionization of HOMO-1. As shown in Fig. 2(b) in the main text, the wave-packet

evolves very rapidly as the changes of θ_{DOD} and its excited energy decreases, thus the resonance conditions for the dipole transition from A state to X state can be met at the angle around 120° (see Fig.R1(c)).

Fig. R1. The potential energy curves of ground and excited states of D_2O^+ are presented in (a) and (b), and the comparison between their excited energies and photon energy is shown in (c).

Second, the probability of the dipole transition also depends on the relative angle between the direction of the transition dipole moment and the polarization of the laser field. For the transition from A state to X state, we calculated the orientation-dependent ionization rate with the molecular-orbital Ammosov-Delone-Krainov (MO-ADK model) based on the orbital symmetry of HOMO-1, and found that the O- D_2 axis of ionic molecule prefers to align to the direction of laser direction. As shown in Fig. R2, the calculated distributions of HOMO and HOMO-1 suggest the transient dipole moment direction is perpendicular to the O- D_2 axis of water (i.e., laser polarization direction), thus the probability of the transition from A to X state is also expected to be small ($\mathbf{E} \cdot \mathbf{D} = |\mathbf{E}| |\mathbf{D}| \cos(\theta)$). Based on these two arguments, we believe that the coupling between X and A states is qualitatively less significant in our measurement.

Fig. R2. (a) and (b) are the HOMO and HOMO-1 of D_2O molecule, their transition dipole moment is perpendicular to the molecular plane. (c) and (d) are the corresponding angular-dependent ionization yields distribution from these two orbitals, where θ is the angle between laser polarization and O- D_2 axis.

Furthermore, we conducted quantitative simulations of the coupling effect between X and A states. We calculated the time-resolved population transfers between X and A states through dipole transition. From Fig.R3 (a), we can see that the population transfer from X to A state through dipole transition is very weak even if the laser polarization direction is the same as the direction of the dipole moment (0°). The integrated results for all the orientations shown in Fig. R3(c) are within 10%. For the transition from A state to X state, resonance transition can occur when the nuclear wave-packet evolves for a certain time, which leads to large portions of populations transfer from the A state to the X state (see Fig. R3(b)). However, considering the molecular orientation-dependent strong field ionization yield with MO-ADK model, the integrated results for all the orientations shown in Fig. R3(d) indicate that about 10% of initial populations undergo transitions from A to X state. Importantly, our simulation shows that 10% of population transfer can be only achieved after interacting with a laser pulse for more than 20 femtoseconds. However, for the TERCE, the most probable evolution time of molecules in the cation is generally less than 10 femtoseconds and the coupling between X and A states will be less than 5%.

Fig. R3. Simulated time-resolved population transfers between X and A states through the dipole transition process. (a) shows the results as the X state is initially populated. The time-resolved distributions of populations rely on the angle between laser polarization and dipole moment (0° to 90°). (b) is the same as (a) but the A state is initially populated. (c) and (d) are the results of integrating all the orientations. Here, the molecular orientation-dependent tunneling ionization yields are considered for both HOMO and HOMO-1 ionization. The time axis stands for the interaction time between the molecule and the laser pulse.

Fig. R4. Simulated time-dependent vibrational wave-packets dynamics of X state where the dipole transition from X to A states is considered. The first and second rows from the bottom stand for the wave-packets distribution of X state and A state, respectively, after interaction with laser pulse from 0 to 6 optical cycles. The intensities of wave-packets of A state after dipole transition are enlarged 10 times.

Fig. R5. Simulated time-dependent vibrational wave-packets dynamics of A state where the dipole transition from A to X state is considered. The first and second rows from the bottom stand for the wave-packets distribution of A state and X state, respectively, after interaction with laser pulse from 0 to 6 optical cycles. The intensities of wave-packets of X state after transition are enlarged 10 times.

Finally, we simulated the time-dependent vibrational wave-packets distributions of X and A states when the coupling effects between them are considered. Here, only the dipole transition occurs at the peak of the laser pulse is calculated due to the limitation of the computation resource. When the X state is initially populated (Fig. R4), the coupling from X to A state slightly changes the time-dependent wave-packets distributions of the X state, relative to that without considering the coupling effect (see Supplementary Figure 7). New distributions appear at relative larger θ_{cm} region, which is 1000 times weaker than the dominant distribution of X state. When the A state is initially populated (Fig. R5), the relatively intense dipole transition from A to X state clearly changes the distributions of wave-packets of A state as the evolution time is larger than 5 cycles, relative to that without considering the coupling effect (see Supplementary Figure 8). However, the probabilities of the coupling between two electronic states are overall more than 10 times

weaker than the initial population of both states. Based on those calculated wave-packets, we simulated the (θ_{cm} - KER) distribution of TERCE by including the coupling effect between X and A states in our model, and the result is slightly different relative to that neglecting the coupling effect between electronic states (see Fig. R6), especially, a weak enhancement can be seen in the larger θ_{cm} region (larger than 170°), but the four components we discussed in the main text are still dominant.

Fig. R6. (a) Comparison between simulated (θ_{cm} - KER) distribution of TERCE after considering the coupling effect between X and A states with that (b) neglecting the coupling effect between X and A states.

In summary, the coupling of two electronic states in the laser field can play a role in the strong field multiple ionization, the significance of this effect depends on the detail features of the specific electronic states, such as the excited energy and symmetry of the orbital. In the present case of water, the coupling effect between X and A states is not significant, but it can contribute to the distribution of TERCE in larger θ_{cm} region (larger than 170°). Moreover, since the most probable interaction time between X and A states and the laser field is found to be very short (within 10 fs) in our TERCE approach, this coupling effect is rather weak relative to the direct population via tunneling. Thus in our present work, we focused on the dominant nuclear dynamics triggered by the tunnelling ionization. In the future, fully including the coupling effect between electronic states will be very helpful to make the simulation approach more complete, and we may reveal the interesting nuclear dynamics triggered by the resonance photo-excitation.

Changes: We added all the discussions about the coupling of the X and A states in section IV of the supplementary. Fig. 5 and the related discussions in the main text have been revised to make the statements rigorous and prevent any over-interpretation, and we expressed clearly that the presented time-resolved structural evolutions along the cationic states are obtained based on the simulations. In the abstract, we added "in the present condition" in the fifth sentence to make the statement more rigorous.

Would the temporal evolution be modified if a shorter pulse (say 20 fs or less) is applied?

Reply: we implemented new experiments with different pulse durations of 30 fs and 80 fs, which are available in our laboratory. The distributions from TERCE are shown in Fig. R7. The results of Fig. 1 (a) and (c) in the main text is also inserted for comparison. We simulated the distributions from TERCE and the results are shown in Fig. R7 (g)-(l). We can see the measured distributions present overall similar patterns and rather weak pulse duration-dependence can be observed from 30 fs to 80 fs. The simulated results show consistent features and the dominant components exhibit a similar distribution of KER and θ_{cm} for those three cases (Fig. R7 (g)-(l)), thus the temporal evolution almost stays identical for the pulse duration from 30 fs to 80 fs. The agreement between the measurements and simulations also can verify the validity of our simulation approach.

For the laser pulse duration shorter than 20 fs, we simulated the (θ_{cm} - KER) distributions from TERCE, and the results are presented in Fig. R8. The distributions of θ_{cm} shrink to smaller angles as the pulse duration decrease from 20 fs to 5 fs. As the pulse duration decreases to 5 fs, the distribution with large θ_{cm} is disappear, and the results are dominant by the two components at small θ_{cm} (smaller than 150°). This is because the evolution of wave-packets in the cationic X and A state are nearly frozen, and the wave-packets have been directly projected to the dication before any prominent bending motions for the A state. For such a short laser pulse, the rapidly decreasing electric field does not allow molecules to ionize after a long period of evolution in both the X and A states. In fact, the simulation suggests that tunneling ionization mainly occurs within the strongest cycle of the laser pulse. Since the laser system with a pulse duration shorter than 10 fs is not available in our laboratory, the experimental distribution of TERCE can not be obtained and compared with the simulation. However, in Fig. R8(d), we present the (θ_{cm} -KER) distribution of the three-body Coulomb explosion triggered by the electron impact-induced vertical triple ionization. In this experiment, the evolution of wave-packet on the cation and dication states is also fully frozen. Relative to the case of TERCE with 5 fs laser pulse (Fig. R8(c)), their distributions of θ_{cm} are comparable. We found the peak position (θ_{cm}) of the two components in Fig. R8(c) (125° , 144°) and Fig. R8(d) (122° , 144°) are consistent in a quantitative level (see Fig. R9). For the distributions of KER, the peak value in (c) is smaller than (d), which originates from the bond stretching motion along the dication state during the electron re-collision process (2 fs). Those results suggest the temporal evolution of the wave-packet along the cation can be affected by the pulse durations. The comparison between the simulations of TERCE (Fig. R8(c)) and the measurement with electron impact ionization can also verify the validity of our simulation approach.

Fig. R7. (θ_{cm} -KER) distributions of the measurements with different pulse durations (a) to (c) and their selected events mainly from TERCE (d) to (f). The total counts in (c) are less than other conditions. (g) to (i) are the simulated (θ_{cm} -KER) distributions with the same pulse durations as the measurements.

Fig. R8. (a) - (c) Simulated (θ_{cm} -KER) distribution of TERCE for the pulse duration of 20 fs, 10 fs, and 5 fs. (d) presents the experimental three-body CE results induced by the electron impact-induced vertical triple ionization. The marked distributions with black circles indicate two CE pathways from the neutral ground state to the two states of trication.

Fig. R9. One-dimensional distributions of θ_{cm} for the two dominant components (KER: 30 eV - 39 eV) in the electron impact measurement (black balls) and the simulated result of TERCE with pulse duration of 5 fs (pink solid line).

Changes: We added the comparison of measured and simulated TERCE with pulse duration of 30 fs, 40 fs, and 80 fs in Supplementary Figure16. The simulations with pulse durations of 5 fs to 20 fs and the comparison with electron impact measurement are also added in the Supplementary Figure 17 in the supplementary information.

2. In the NSDI calculation, the so-called MCTDW model was used to calculate the electron impact ionization probability. This model is based on the distorted-wave approximation; thus, it is not expected to work for electron impact on molecular ions and for low energy electrons. Since the returning electrons have a range of energies, integration over the returning electron energy is required. Has this been done?

Reply: We thank the reviewer for the helpful comments and we fully agree with the reviewer that integration over the returning electron energy is necessary. Indeed, the MCTDW is a model based on the distorted-wave method. In the present calculations, we have considered the influence of direct distorted potential, polarization potential, and exchange potential in the distorted-wave calculation. Moreover, we also considered the effect of the multicenter nature of molecules on the continuous electron wave function. With these developments, the MCTDW method has been proven to be capable to simulate the low-energy electron impact ionization probability of molecular ions [see details in Phys. Rev. A 104, 012817 (2021), J.Phys. B. 54,015206 (2021), Phys. Rev. A.105.042805 (2022)], especially for water molecule [Phys. Rev. A 98, 042710 (2018)].

In the previous simulation of TERCE, we did not consider the energy distribution of the returning electrons. Hence, we perform a new simulation that includes the contributions from the returning electrons with different energies. In this simulation, we first calculated the energy distribution of the returning electrons when the water molecule is ionized at different instants of each optical cycle for the laser pulse, and then calculated cross sections of electron impact ionization with different electron energies. The obtained total cross-sections as a function of electron energy have been shown in Fig. R11(a). The results show that the total cross-sections decrease very slowly as increasing the incident energy. The calculated final distributions of TERCE are presented in Fig. R10(a), and (b) the distribution without considering this effect is also presented for the comparison.

Fig. R10. (a) and (b) presents the simulated (θ_{cm} -KER) distribution where the different cross sections for the returning electrons with different energies are considered (a) or not considered(b).

Changes: The new results considering the energy integration of the returning electrons is updated in the new figure 1(e) of the main text.

3. For NSDI, it is well-known that electron impact excitation could be more important than electron impact ionization. Since the excited electron is readily ionized in later cycles, so impact excitation is also treated as NSDI. This possibility was not addressed and it would change the conclusion of the simulation.

Reply: As the reviewer pointed out, the total cross-section of electron impact excitation is larger than the cross-section of electron impact ionization in the low energy region. We have calculated the total cross section for the electron impact excitation of D_2O^{2+} from ground state to the first excited state (see Fig. R11(a)), it is indeed higher than that of electron impact ionization, but they are on the same order of magnitude. However, the contribution of electron impact excitation to the TERCE also depends on the tunneling ionization rate of excited state (D_2O^{2+*}) to D_2O^{3+} . Due to the high ionization energy of D_2O^{2+*} in the small bond length region, the tunneling ionization rate of D_2O^{2+*} is extremely low (in the order of 10^{-9} , see Fig. 11(b)), and thus the final contributions of the electron impact excitation to the TERCE should be very small relative to the one by direct electron impact ionization. Thus although the electron impact excitation has larger total cross sections than electron impact ionization, its contribution to the TERCE is expected to be negligible.

Fig.R11. (a) Calculated the total cross sections of electron impact excitation and electron impact ionization. (b) presents the ionization probability of dication after the electron impact excitation.

Changes: We added Supplementary Figure 6 to address the contributions of electron

impact excitation to the TERCE in the Supplementary information.

There are other issues in their “nine approximations” and others addressed above. Their simulation model has been much more detailed than what is available so far for retrieving dynamics from CEI data. Granted that these approximations might not “hurt” the dissociation dynamics extracted from their simulations. On the other hand, to establish the validity of the present simulation model, the best test is to apply the simulation using different laser parameters, to see if different temporal dynamics (similar to Fig. 5) can be found and then to do CEI experiments with those laser parameters. **If this is carried out, then the validity of the approximations used in the simulation is established, and the retrieved molecular dynamics versus time can then be more believable.**

Reply: In the new version of the manuscript, we added new simulations of TERCE covering pulse durations from 5 fs to 80 fs. We compared the simulation with experimental results for the pulse duration of 80 fs, 40 fs, and 30 fs. For the shortest pulse duration of 5 fs, we found the distributions of the simulated θ_{cm} become narrower and are all below 150° , and their peak values of two dominant components are quantitatively consistent with the distribution from three-body Coulomb explosion results triggered by the electron impact-induced vertical triple ionization. Those comparisons between simulations and measurements show reasonable agreements, which provide new evidences to validate the simulation approach.

Moreover, we further provide more discussions to justify the approximations used in the simulations. In the current approach, we simulated the nuclear three-body Coulomb explosion processes for the sequential triple ionization from D_2O to D_2O^{3+} . Such processes are very complex and include three different types of dynamics: (1) the evolution of nuclear wave-packets in different electronic states, (2) the strong field tunneling ionization and (3) the impact ionization from dication to trication by laser-induced returning electrons. The theoretical developments of these three directions are usually independent and have not been combined to simulate the molecular breakup dynamics. When we integrate those theoretical approaches together, we need to inherit their approximations in each direction, and at the same time, we also need to add some new approximations to enable the combination of those three theoretical approaches. As a result, nine approximations have to be included in present simulations. According to the review’s comments, we further try to provide the rationality of each approximation in more details.

(1): The ionization by laser and electron collision both satisfy the Franck-Condon principle. This is one of the most widely used approximations in the simulations of both strong field ionization and electron impact ionization.

(2): The laser field can be considered as a perturbation interaction, and only affects the distribution of orientation. Its influences on the coupling of the vibrational states and electronic states are ignored. The approximation is justified as following: (i) The vibration

time scale of molecules is usually significantly larger than the period of the laser optical, resulting in the cancellation of the work done by the oscillating fields to the molecular vibration. (ii) For the dipole transition from X to the A state via absorbing photon, we calculated the excited energy between X and A states. The energy difference between X and A states (D_2O^+) is 0.7 eV larger than the photon energy (1.55 eV for 800 nm pulse) (see Fig. R1), thus the resonance condition of dipole transition can not be met, and the probability of transition from X to A state is expected to be small. For the dipole transition from A state to X state via emitting photon, the orientation-dependent ionization rate shows that the O- D_2 axis of ionic molecule prefers to align to the direction of laser polarization direction, which is perpendicular to the transient dipole moment direction. Hence, the probability of the transition from A to X state is also not significant. More importantly, in the TERCE, the most probable interaction time between laser and X (or A) states is within 10 fs, which leads probability of the coupling of X and A states to be less than 5% relative to the initial populations in the present experimental condition. Finally, we simulated the (θ_{cm} -KER) distribution from TERCE when the coupling effects between X and A states are included. We found this coupling effect can play a minor role in TERCE by enhancing the distribution in the larger θ_{cm} region (larger than 170°), however, the contributions from the direct tunnelling ionization are still dominant. Thus we mainly focused on the nuclear dynamics triggered by the directly tunnelling ionization in the present manuscript, and will explore the interesting nuclear dynamics triggered by the photo-excitation in the future.

(3): The MO-ADK method is used to calculate the ionization yields of D_2O . This method is an established way to calculate the probability of strong field ionization of molecule where the symmetry of the orbital can be considered. To the best of our knowledge, this is one of the most recognized methods to calculate the strong field ionization yield of simple molecules.

(4): The electron-water collision ionization cross sections are calculated by the MCTDW method. Here, the influence of direct distorted potential, polarization potential and exchange potential in the distorted-wave calculation have been considered. Moreover, the effect of the multicenter nature of molecules on the continuous electron wave function is also considered. With these developments, the MCTDW method has been proven to be capable to simulate the low-energy electron impact ionization probability of water ion [Phys. Rev. A 98, 042710 (2018)].

(5): The average of the molecular wave-packets is treated as the molecular configurations at different ionization times. This approximation is also frequently used in the simulation of strong field ionization and electron impact ionization.

(6): Only the ionization at the peak of the electric field in each laser cycle is considered in the simulations. This approximation is reasonable since the strong field ionization yield exponentially depends on the amplitude of the electric field for the strong field tunneling ionization.

(7): A semi-classical trajectory method is used to simulate the three-body Coulomb

explosion of D_2O^{3+} . This method is widely used to simulate the Coulomb explosion of highly charged molecular ions. Its validity is also justified by well reproducing our (θ_{cm} -KER) distributions from direct triple ionization induced by the electron impact ionization.

(8): The influence of intensity volume effect of strong field ionization on Coulomb explosion imaging is ignored because of the limitation of computing resources. This approximation has been verified by the similar results obtained at different laser intensities.

(9): Considering the evolution of rotating wave-packet before and after ionization through equation (5). This approximation is used to connect the three different theoretical approaches.

Changes: We revised the summary of all the approximations used in this simulation, and added more detailed justifications for the approximations. Moreover, the discussions about the coupling of electronic states and Supplementary Figure 4 - 6, Figure 16-17 are added in the supplementary. The Fig. 5 in the main text is revised to make the statements more precisely.

In summary, despite that the authors have addressed what are the approximations made in their simulation, it is still “dangerous” not to worry that their simulation is not good enough. To make the claim that the molecular dynamics extracted from the CEI data from their simulation would require additional tests from additional experiments. To claim a “big” progress on a long-standing problem requires higher standard. Such a big claim has been made in this manuscript, but more evidence is needed. In its present form, I have difficulty to recommend its publication in Nature Communications.

Reply: As the reviewer suggested, we did additional TERCE experiments with available laser pulse durations in our laboratory, and implemented new simulations covering broad range of pulse durations from 5 fs to 80 fs. Both the measurements and simulations with pulse durations of 30 fs, 40 fs, and 80 fs show similar (θ_{cm} -KER) distributions from TERCE, which verifies the validity of our simulations. As pulse duration reduces from 20 fs to 5 fs, the simulated distributions of TERCE shrink gradually to the small θ_{cm} , since the bending motion in the A state of cation is frozen by the ultrashort laser pulse. For the shortest pulse duration of 5 fs, the simulation can mimic the θ_{cm} distribution of the vertical triple ionization induced by the electron impact, where the nuclear motion along the cation is also frozen. The quantitative agreement of θ_{cm} between them can be reached, which can further support the validation of our simulations. We agree with the reviewer that the temporal dynamics rely on the pulse durations, thus we revised Fig. 5 and the correlated discussions in the main text to summarize the results more precisely. By adding new experimental evidences and simulations, we believe that the extracted electronic states-resolved transient structures of water molecule induced by valence and inner-shell ionization are solid in the present experimental conditions.

Comments to the reply to reviewer 1.

Please refer to the page number of the rebuttal file submitted by the authors. Important comments are in boldface.

Page 2.

Comments on CEI measurements by 200 eV electrons are irrelevant. Triple ionization in this case can be due to electron impact to autoionization state of dication (followed by Auger) or by direct triple ionization. These processes are different from the triple ionization by D2O by intense laser discussed in the paper. Their answer to this question is misleading and cannot be correct. I also do not see how you can calculate triple ionization and what autoionizing states were included in the simulation. This answer makes me worry about the "simulations" reported in this MS.

Reply: We are sorry for the previous misleading reply. For the three-body Coulomb explosion induced by 200 eV electron impact, the interaction time of electron collisions is in an attosecond regime, and the neutral molecules are directly populated into highly charged states, thus its simulations are different from that used for the simulation of TERCE. However, for the breakups of D_2O^{3+} , the same semi-classical trajectory method is used to simulate the momentums of D^+ and O^+ for both electron impact ionization and TERCE. Considering that the energy of the incident electron reaches 200 eV, the ground state and several excited states of trication are included in the simulation of electron impact ionization. For the breakups of the autoionization state, the calculated orbital ionization energy shows that only the D_2O^{2+} ($2a_1^{-2}$) state can contribute to the three-body breakups with KER ~ 30 eV. Since the potential energy surface of D_2O^{2+} ($2a_1^{-2}$) is also strongly repulsive, the semi-classical trajectory method is also suitable to simulate its breakup. We simulated the (θ_{cm} -KER) distribution of fragments after autoionization to trication at different times. Finally, by comparing the simulated KER with the experimentally measured value, we can obtain the time of autoionization and also the values of θ_{cm} .

Changes: We provide the details of the simulations for the three-body breakups originating from the electron impact-induced vertical triple ionization and autoionization in the supplementary.

Page 3 to page 6.

Thanks for explaining the details of how the data are analyzed and for clarifying that Region II and III are about 100 times smaller than region 1.

Reply: Thanks to provide valuable comments

Bottom of page 6

It would be essential to address what is the energy range of the rescattering electrons. In calculating the ionization rate for NSDI, one has to integrate over the energies of the rescattering electrons. How these are done and how the molecular orientation is included?

Rescattering excitations are often more important than rescattering ionization for NSDI. The excited electron in general will be ionized at subsequent laser cycles. How to justify it does not contribute to the NSDI?

Reply: Please refer to the reply to question 2.

Page 7

The statement, "Due to the limitation of computing resources...", is not an excuse for not answering my question. Why rotation is so important in a laser field for time scale of a few femtoseconds? I would expect that at such short time scale, the electronic dynamics is more likely modified by the laser. Since the authors do not account for or examine laser couplings among the electronic states, it is not clear how to justify treating the "evolution" of each excited electronic states as independent. This is actually a critical assumption that should be addressed.

Reply: We are sorry for the misunderstanding of your previous comment about the orientation effect. For the coupling effects of electronic states in the laser field, please refer to our reply to question 1. Here, we explain how we consider the orientation effect in the multiple ionization and dissociation dynamics in TERCE in our approach.

For the initial tunneling ionization of neutral water, the MO-ADK model is used to estimate the tunneling ionization rate, and the different initial molecular orientations are considered. Molecular orientation can affect the relative tunneling yields ratio between X and A states, but has little influence on the vibrational wave-packets evolution. Similarly, the orientation effect of tunneling ionization from cation to dication is also included by the MO-ADK model. For molecules in a strong laser field, the interaction between laser pulse and molecules can be written as $\mathbf{E} \cdot \mathbf{D} = |\mathbf{E}| |\mathbf{D}| \cos(\theta)$, where \mathbf{E} is the electric field of laser, \mathbf{D} is the dipole moment and θ is the angle between molecular axis and laser polarization direction. According to this formula, all laser-molecule interactions will change the molecular orientation. For the neutral molecule, the laser-induced polarization effect of molecule can lead to the change of its rotational wave function, which is the second-order perturbation term of $\mathbf{E} \cdot \mathbf{D}$ and proportional to the intensity of the laser pulse. If the intensity of the laser pulse can reach to the level of 100 TW/cm², the polarization effect can lead to the change of molecular potential energy in the order of ~eV, which will rotate the axis with the highest molecular polarizability to the direction of laser polarization within tens of femtoseconds. However, for the D₂O molecule, its dipole polarizabilities in the three axes are very close (O-D₂: 9.61, D-D: 9.38, Vertical molecular plane: 9.96), thus the interaction is very weak. Furthermore, the coupling of electronic states is also correlated with the orientation of molecules since the dipole transitions interaction is still $\mathbf{E} \cdot \mathbf{D}$, as we discussed in the reply to question 1. Finally, for the electron-recollision induced ionization from dication to trication, the orientation effect is considered by MCTDW method, in which the directions of electron emission and rescattering are set to the polarization direction of the laser pulse.

Changes: We added the related discussions about the molecular orientation effects in the Supplementary information.

REVIEWERS' COMMENTS

Reviewer #1 (Remarks to the Author):

Third report

The authors have addressed many of the questions I raised in my 2nd report and performed additional experiments using pulses of different durations, and further simulations, to support their conclusions. At this point, I would suggest that the manuscript be accepted for publication in Nature Communications, with the condition that they tone down the claim made in their paper.

We recall what they have done in this paper. A typical intense 800 nm laser was used to hit the D₂O molecule up to triply ionized molecular ions. By measuring the correlation of angle and kinetic energy release (θ , KER) of the three-body Coulomb explosion products, with linearly and circularly polarized laser, the authors claimed that they were able to "Directly imaging the excited state-resolved transient structure of water" with few picometers and few femtosecond resolutions. As reported in Fig. 5, based on their simulations and experimental results, they claimed that they were able to obtain the time dependence of how the molecule/molecular ions break. Such a conclusion is based on theoretically calculated (θ , KER) spectra that are in reasonable agreement with spectra obtained in the experiment. However, retrieving the temporal and spatial properties from the measured spectra is an inverse scattering problem that do not have unique solutions. Even assuming that the experimental data are exact, the retrieved temporal-spatial results at intermediate time steps would have inherent uncertainty. Unless thousands of similar calculations are carried out, such uncertainty cannot be revealed. The latter is of course impossible because excessive computations will be required.

Trying not to be unfair and not too critical of the authors, I decided to recommend that this article be accepted for publication in Nature Communications with one condition since such practice is prevalent in the literature today. The condition is that the authors should state that their claimed results are limited by the accuracy of theoretical simulations. Such statements should be made in the abstract and in the conclusions, thus leaving room for further future experimentation and simulations.

Reviewer #1 (Remarks to the Author):

Third report

The authors have addressed many of the questions I raised in my 2nd report and performed additional experiments using pulses of different durations, and further simulations, to support their conclusions. At this point, I would suggest that the manuscript be accepted for publication in Nature Communications, with the condition that they tone down the claim made in their paper.

We recall what they have done in this paper. A typical intense 800 nm laser was used to hit the D₂O molecule up to triply ionized molecular ions. By measuring the correlation of angle and kinetic energy release (Θ , KER) of the three-body Coulomb explosion products, with linearly and circularly polarized laser, the authors claimed that they were able to “Directly imaging the excited state-resolved transient structure of water” with few picometers and few femtosecond resolutions. As reported in Fig. 5, based on their simulations and experimental results, they claimed that they were able to obtain the time dependence of how the molecule/molecular ions break. Such a conclusion is based on theoretically calculated (Θ , KER) spectra that are in reasonable agreement with spectra obtained in the experiment. However, retrieving the temporal and spatial properties from the measured spectra is an inverse scattering problem that do not have unique solutions. Even assuming that the experimental data are exact, the retrieved temporal-spatial results at intermediate time steps would have inherent uncertainty. Unless thousands of similar calculations are carried out, such uncertainty cannot be revealed. The latter is of course impossible because excessive computations will be required. Trying not to be unfair and not too critical of the authors, I decided to recommend that this article be accepted for publication in Nature Communications with one condition since such practice is prevalent in the literature today. **The condition is that the authors should state that their claimed results are limited by the accuracy of theoretical simulations. Such statements should be made in the abstract and in the conclusions, thus leaving room for further future experimentation and simulations.**

Reply: We thank the reviewer for the valuable comments and suggestions for the acceptance of our manuscript. We agree with the reviewer that the accuracy of our results is limited by the theoretical simulations, thus we added the sentence “The accuracies of the results are limited by the simulations.” both in the abstract, introduction and the conclusion, as the reviewer suggested. We are grateful for the reviewer’s thoughtful comments for the last three rounds, which have indeed significantly improved our manuscript.